# RNAi for Western Corn Rootworm Management: Lessons Learned, Challenges, and Future Directions

**DOI:** 10.3390/insects13010057

**Published:** 2022-01-05

**Authors:** Molly Darlington, Jordan D. Reinders, Amit Sethi, Albert L. Lu, Partha Ramaseshadri, Joshua R. Fischer, Chad J. Boeckman, Jay S. Petrick, Jason M. Roper, Kenneth E. Narva, Ana M. Vélez

**Affiliations:** 1Department of Entomology, University of Nebraska, Lincoln, NE 68583, USA; mndarlington@gmail.com (M.D.); jordan.reinders3@gmail.com (J.D.R.); 2Corteva Agriscience, Johnston, IA 50131, USA; amit.sethi@corteva.com (A.S.); albert.l.lu@corteva.com (A.L.L.); chad.boeckman@corteva.com (C.J.B.); jason.roper-1@corteva.com (J.M.R.); 3Bayer Crop Science, Chesterfield, MO 63017, USA; partha.ramaseshadri@bayer.com (P.R.); joshua.fischer@bayer.com (J.R.F.); jay.petrick@bayer.com (J.S.P.); 4GreenLight Biosciences, Research Triangle Park, NC 27709, USA; knarva@greenlightbio.com

**Keywords:** RNAi, western corn rootworm, *Diabrotica virgifera virgifera*, plant-incorporated protectant, pyramid strategy, insect resistance management

## Abstract

**Simple Summary:**

The western corn rootworm (WCR), *Diabrotica virgifera virgifera* LeConte, is an annual pest of maize in the United States Corn Belt. Larval feeding on the root system can promote significant yield loss through reduced water and nutrient uptake and decreased plant stability. Various management tactics, including crop rotation, insecticides, and transgenic crops expressing *Bacillus thuringiensis* Berliner proteins, have been used to manage WCR densities. However, resistance has evolved to each of these tactics in local areas, highlighting the need for new management strategies. The use of RNA interference (RNAi) technology for WCR management represents the next phase of species-specific pest management. This paper reviews the current knowledge of RNAi for WCR management. We present an overview of traits that have been explored and the accumulated knowledge acquired on mode of action, resistance, ecological risk assessment, and mammalian safety. We conclude by highlighting the challenges and future directions of this technology for WCR management.

**Abstract:**

The western corn rootworm (WCR), *Diabrotica virgifera virgifera* LeConte, is considered one of the most economically important pests of maize (*Zea mays* L.) in the United States (U.S.) Corn Belt with costs of management and yield losses exceeding USD ~1–2 billion annually. WCR management has proven challenging given the ability of this insect to evolve resistance to multiple management strategies including synthetic insecticides, cultural practices, and plant-incorporated protectants, generating a constant need to develop new management tools. One of the most recent developments is maize expressing double-stranded hairpin RNA structures targeting housekeeping genes, which triggers an RNA interference (RNAi) response and eventually leads to insect death. Following the first description of *in planta* RNAi in 2007, traits targeting multiple genes have been explored. In June 2017, the U.S. Environmental Protection Agency approved the first *in planta* RNAi product against insects for commercial use. This product expresses a dsRNA targeting the WCR *snf7* gene in combination with *Bt* proteins (Cry3Bb1 and Cry34Ab1/Cry35Ab1) to improve trait durability and will be introduced for commercial use in 2022.

## 1. Introduction

The western corn rootworm (WCR), *Diabrotica virgifera virgifera* LeConte (Coleoptera: Chrysomelidae), is one of the most destructive insect pests of maize (*Zea mays* L.) in the United States (U.S.) [1,2]. Native to Central America and first identified as a pest of cultivated maize in Colorado in 1909 [3,4], populations are now found throughout the midwestern U.S. and Europe [5,6]. Once achieving pest status, various control tactics have been used to reduce damage caused by WCR larval feeding including crop rotation, soil-applied insecticides, and maize hybrids expressing insecticidal proteins from the soil bacterium *Bacillus thuringiensis* Berliner (*Bt*) [2,7,8]. However, managing WCR has been historically challenging due to its remarkable ability to evolve resistance to all available management tactics throughout various local areas in the U.S. Corn Belt [2,7,9,10,11,12,13,14]. Economic analyses estimate costs associated with control strategies and yield loss exceed USD $2 billion annually [1,15]. Four insecticidal *Bt* proteins are currently available for WCR management: Cry3Bb1, mCry3A, eCry3.1Ab, and Cry34Ab1/Cry35Ab1 (now classified as Gpp34Ab1/Tpp35Ab1 [16]) [17,18,19,20]. Field-evolved resistance to Cry3Bb1 and mCry3A was first reported in 2011 [21] and has subsequently been confirmed in various areas of the U.S. Corn Belt [22,23,24,25,26,27]. Cross-resistance between Cry3Bb1, mCry3A, and eCry3.1Ab has been widely demonstrated [21,23,25,26,28]. More recently, resistance to Cry34Ab1/Cry35Ab1 was identified [27,29,30], highlighting the urgency for alternative approaches for WCR management. Recent review articles highlight the history, use of, and evolution of resistance to synthetic insecticides [7] and *Bt* traits [8].

*In planta* expression of double-stranded RNA (dsRNA) in the shape of a hairpin RNA triggers an RNA interference (RNAi) response within the insect, representing a new mode of action for WCR control [31,32]. In June 2017, the U.S. Environmental Protection Agency (EPA) registered the first transgenic maize product with an RNAi-based plant-incorporated protectant (PIP) for WCR management [33]. This product expresses three *Bt* proteins (Cry3Bb1 and Cry34Ab1/Cry35Ab1) and a dsRNA [32]. This review summarizes our current knowledge of RNAi for WCR management, including studies on the various RNAi traits explored to date, mode of action, susceptibility of field populations, resistance evolution, environmental risk assessment, and mammalian safety.

## 2. RNAi in the Western Corn Rootworm

RNAi is an inherent post-transcriptional gene silencing mechanism utilizing various non-coding RNAs (ncRNA) as substrates [34]. RNAi is thought to have evolved as a defense mechanism against invading nucleic acids such as viruses and transposable elements or anomalous endogenous transcription products [35,36]. RNAi was first described in *Caenorhabditis elegans* in 1998 and was subsequently shown to be conserved in most eukaryotic organisms [37,38].

Three RNAi pathways are present within eukaryote organisms, each triggered by distinct, short, duplexed, non-coding nucleic acids: microRNA (miRNA), piwi-interacting RNA (piRNA), and short interfering RNA (siRNA) [39] (reviewed by Zhu and Palli [40]). Each ncRNA embeds within an Argonaute protein and acts as a guide for the direct targeting of a single-stranded RNA molecule within the cell [41]. Despite overlap within the core machinery, each pathway differs in functionality. The RNAi pathway triggered by miRNA is involved in miRNA processing [42], regulation of nucleic acids through translation repression [43,44], and target degradation [45,46,47,48]. The piRNA pathway is involved in the silencing of transposable elements [49] and germline development [50]. Finally, the siRNA pathway responds to endogenous siRNAs [51] and long exogenous dsRNAs present during viral infections [52], which are then sliced into siRNAs within the cytoplasm. The siRNA pathway can be triggered exogenously through the application of synthetic dsRNA for specific silencing of inherent messenger RNA (mRNA) [34]. This pathway is commonly used for gene function studies and, more recently, for pest management [53,54]. Thus, this review focuses on the siRNA RNAi pathway in WCR.

In 2007, Baum et al. [55] demonstrated that oral exposure to *DvV**-ATPase A* dsRNA elicited a silencing response in WCR larvae, leading to mortality (LC_50_ < 0.52 ng/cm^2^). *In planta* expression conferred maize root protection (mean node injury score of 0.25), exhibiting the possibility of utilizing the natural RNAi response for pest management. Another early study used RNAi in WCR to assess the function of WCR orthologues to *laccase 2* (*lac2*) and *chitin synthase 2* (*chs2*) by injecting dsRNA into second and third instars [56]. Injection of 200 ng/µL *DvLac2*-specific dsRNA resulted in a 95.8% reduction in *lac2* expression and the prevention of post-molt cuticular tanning, while injection of 200 ng/µL *DvCHS2*-specific dsRNA reduced chitin levels in midgut tissues by 78%. Both studies demonstrated that WCR has a robust RNAi response and systemic gene knockdown could be generated in WCR through both injection and feeding.

Silencing via the siRNA pathway involves three steps: dsRNA uptake, gene silencing, and systemic spread. Understanding the RNAi pathway in WCR is imperative to characterize potential resistance mechanisms. Much of the mode of action in WCR is understood and will be described in further detail in this review. However, knowledge gaps still exist and further examination in WCR is warranted.

### 2.1. dsRNA Uptake

To elicit an RNAi response, dsRNA must first enter the cell. In WCR, current evidence points to two putative uptake mechanisms: systemic RNA interference defective-like proteins (SID-like or SIL) and clathrin-mediated endocytosis (CME). In *C. elegans,* various SID proteins facilitate dsRNA movement across biological membranes [57,58,59,60,61]. Coleopteran homologs of *C. elegans sid* genes are labeled *sid-like* (*sil*) due to more substantial homology with *tag-130/chup-1*, a protein involved in cellular cholesterol import, than with SID proteins themselves [62]. At least two *sils* are found within the WCR genome (GenBank: PXJM00000000.2), *silA* (LOC114327414) and *silC* (LOC114340333). In WCR larvae, knockdown of each gene individually reduced the phenotype of a second dsRNA application (targeting *DvEbony*), implicating *silA* and *silC* as putative routes for dsRNA uptake in WCR [63]. However, in adult WCR, *silA* silencing did not affect *V**-ATPase* knockdown after secondary exposure to *DvV-ATPase* dsRNA. Despite this, mortality levels typically associated with a *DvV-ATPase* dsRNA exposure were reduced in *silA* knockdown treatments, while silencing of *silC* alone and in combination with *silA* had no effect on *V-ATPase* silencing or mortality [64]. Due to these ambiguous results, more evidence is required to determine if *silA* and *silC* are indeed involved in dsRNA uptake in WCR. In *C. elegans, tag-130/chup-1*, the gene most similar to coleopteran *sils,* plays no role in systemic RNAi [62]; therefore, it is conceivable that *silA* and *silC* may have a function unrelated to dsRNA uptake. In addition, some insects with systemic RNAi responses lack SID homologs within their genome, suggesting the existence of alternative routes of cellular uptake [65].

The clearest evidence for dsRNA uptake in WCR is clathrin-mediated endocytosis (CME) [64]. CME is a form of endocytosis by which macromolecules such as proteins, lipids, and pathogens enter a cell in a clathrin-dependent manner [66,67]. CME requires cell surface receptors to recognize ligands and initiate internalization [68]. In WCR, silencing of key CME factors *clathrin-heavy chain* (*chc*), *vacuolar H^+^ATPase 16 kDa subunit* (*vha16*), and *clathrin adaptor protein AP50 (AP50)* led to a reduction in marker dsRNA effect, while knockdown of *ADP ribosylation factor-like 1* (*Arf72A*) and *small GTPase Rab7* did not produce the same effect [64]. Both AP50 and CHC are required for clathrin-coated pit formation, in which these proteins congregate around the intracellular portion of a receptor and allow for endosomal development [69]. Vha16 is also necessary for endosomal maturation by regulating vesicle acidification [70]. Interestingly, knockdown of *chc* only slightly reduced silencing caused by a marker dsRNA in adult WCR [64], while a similar assay using *T. castaneum* larvae found much a stronger effect on marker gene silencing [71]. The same experiment also identified Rab7 as an essential component in dsRNA uptake in *T. castaneum* [71], while in WCR, *rab7* knockdown did not show result in a reduction in the silencing of the marker gene [64]. Similarly, results from experiments with Colorado potato beetle, *Leptinotarsa decemlineata,* larvae mirrored those of *T. castaneum* [65]. Divergent results between coleopterans, *D. v. virgifera* (Chrysomelidae), *L. decemlineata* (Chrysomelidae), and *T. castaneum* (Tenebrionidae) could be due to physiological differences between species or from variable responses between life stages. Results suggesting involvement of *silA* and *silC* in the RNAi mechanism in *T. castaneum*, *L. decemlineata*, and WCR were found using the larval life stage. However, similar experiments carried out with WCR adults were not definitive. Expression of core RNAi genes was found to change based on the larval stage in *L. decemlineata*, affecting dsRNA efficacy [72] This finding provides an argument for life stage-dependent dsRNA uptake. However, further examinations of CME genes on WCR larvae are warranted to determine if these differences are due to physiological differences between insect species or life stages.

Despite conflicting results, the involvement of CME in dsRNA uptake in insects, and specifically in Coleoptera, is well established [65,70,71,73], suggesting that CME is the primary mechanism for dsRNA uptake in WCR (Figure 1a). Two key aspects of CME uptake, receptors responsible for dsRNA binding and genes involved in endosomal escape, have yet to be examined in WCR. Internalization of dsRNA is a significant barrier to RNAi efficacy in other insects [74]; therefore, understanding how dsRNA enters and is released from the endosome in WCR may illustrate possible deficiencies in other insects [75,76].

### 2.2. Silencing Mechanism

The siRNA pathway is initiated when dsRNA within the cytoplasm is cleaved into multiple siRNAs by the RNase III family ribonuclease Dicer-2 (Dcr-2) [77,78,79]. Dcr-2 and the dsRNA-binding protein R2D2 form a complex to bind and cleave dsRNA [80]. siRNAs are 21–23 bp long duplexes with a 3’ two-nucleotide overhang [81,82,83]. A single copy of *dcr-2* has been annotated in the WCR genome (LOC114339627). Knockdown of *dcr-2* in WCR adults prevented the silencing of a marker dsRNA, indicating the requirement of Dcr-2 in the WCR RNAi response to exogenous dsRNA [84,85]. Knockdown of *dcr-2* in female WCR led to a reduction in oviposition and silencing during the third instar negatively affected adult emergence [86]. However, knockdown of *dcr-2* in neonates did not affect larval growth or development [87]. R2D2 not only plays a role in dsRNA binding but is also vital for loading siRNAs into the RNA-induced silencing complex (RISC) [88,89,90]. One copy of *r2d2*, expressing three isoforms, has been identified in WCR (LOC114342393), and silencing in WCR females led to a reduction in egg-laying capacity [86]. Based on a homology search within the WCR genome, one more *r2d2* gene may exist; however, functional evaluation is lacking.

After cleavage by Dcr-2, siRNAs are loaded onto the RISC by the RISC loading complex (RLC), comprised of Dcr-2, R2D2 [90,91], and the TATA-Box Binding Protein Associated Factor 11 (TAF11) [92]. Each siRNA is individually bound to an RNase H-like endonuclease, Argonaute-2 (Ago-2) [93,94], within the RISC. Argonaute proteins then separate siRNA duplexes into two single-stranded RNAs, a guide and a passenger strand, after which the passenger strand is degraded and the guide is retained [93,95]. Once the RISC/guide-siRNA complex locates a single-stranded RNA target through base-pair matching, silencing commences via mRNA hydrolysis [81,96,97]. Two silencing requirements include a catalytically active Argonaute protein, which does the actual degradation, and a sequence match between the guide siRNA and the target mRNA [98]. Data from *T. castaneum* suggest a greater than 80% sequence identify with target mRNA is required for an efficient RNAi response [97] and in *D. melanogaster*, 19 bp homology with mRNA led to effective target silencing [99]. After initial hydrolyzation of the mRNA target by Argonaute, the remaining nucleic acid is either degraded by endoribonucleases, exoribonucleases, or translated into incomplete proteins [100]. Multiple *ago2*s may be present in the WCR genome; however, current research has focused on one gene (LOC114327218). Knockdown of *ago2* prevented the downregulation of genes targeted by dsRNA, indicating the requirement of AGO2 for the RNAi response in WCR. Knockdown of *ago2* did not affect WCR development or survival [63,85].

The core RNAi silencing mechanism is relatively well conserved, and data from WCR have generally matched that of other insects. However, while key aspects of the pathway have been evaluated in WCR, many putative steps related to dsRNA and siRNA movement within the cytoplasm remain unstudied. For example, in *D. melanogaster*, the molecular chaperone complex Hsc70-Hsp90 facilitates the loading of siRNAs into Ago2 [101], and the small RNA methyltransferase HUA Enhancer 1 (HEN-1) activates the RISC through siRNA methylation [102]. Whether homologs exist with similar functions in WCR is unknown. Furthermore, crosstalk between the siRNA, miRNA, and piRNA RNAi pathways does exist. For example, silencing of *r2d2* and *ago2* negatively impacted miRNA processing in second instar WCR [86]. In *L. decemlineata*, Ago1 (miRNA) and Aubergine (piRNA) are involved in the dsRNA response [103], and the dsRNA binding protein Loquacious participates in both the siRNA and miRNA pathways in *D. melanogaster*, depending on the isoform expressed [104]. The extent to which crosstalk impacts the processing of exogenous dsRNA in WCR has not been determined. Further understanding of crosstalk between RNAi pathways may provide insight into why RNAi is efficient in Coleoptera and could be a cause of the differential response seen across Insecta. Lastly, the identification of coleopteran-specific *staufenC* in *L. decemlineata* opens up the possibility that other coleopterans have specific genes involved in the RNAi mechanisms [105], which may be responsible for enhanced efficacy that is characteristic of Coleoptera. Even though the main components of the dsRNA processing in WCR have been identified (Figure 1a), knowledge gaps remain. While verification of individual components identified in model organisms may be tedious, the strength of the WCR RNAi response is a valuable resource. Characterizing factors conferring the high susceptibility observed may lead to solutions to overcome barriers in other insects.

### 2.3. Systemic Spread

In organisms highly susceptible to RNAi, systemic spread of the silencing signal is an essential step for sufficient silencing. During a viral infection, cells initially exposed to dsRNA reverberate the signal to distal tissues, priming uninfected cells against the pathogen [106,107]. In *C. elegans*, RNA-directed RNA polymerases (RdRp) use primary siRNA and mRNA targets to form secondary siRNAs, amplifying the signal [108,109,110], while SID transmembrane proteins allow dsRNA to move from cell to cell [57,111]. However, putative homologs of SID proteins (SILs) in insects are not required for a systemic RNAi response and most insects lack RdRps entirely [65,112]. As previously stated, at least two homologs of *sid* genes can be found in the WCR genome; however, involvement in the RNAi mechanism is unclear [63,64]. In addition, no RdRp homolog is found in the WCR genome and small RNA sequencing after exposure to dsRNA found no evidence of secondary siRNA production in WCR, which suggests all siRNAs derive from the original dsRNA molecules [113]. However, this sequencing method would not have identified putative 5’ triphosphate modified siRNAs, small ncRNAs shown to be amplified in an AGO2-dependent manner from viral DNA in *D. melanogaster* hemocytes [106]. Whether this RdRp-independent amplification step is present in WCR or if it can be triggered through non-viral DNA is unknown and requires further examination.

Despite a lack of demonstrated causative elements, spread of the RNAi response in WCR has been documented, evidenced by transcript silencing in tissues distal to the exposure site [56,113,114,115]. Recent work in *T. castaneum* and *L. decemlineata* suggests that after the initial dsRNA uptake, siRNAs or dsRNAs travel cell to cell via extracellular vesicles [116,117]. Further work is required to illuminate the pathway of systemic spread in WCR.

## 3. RNAi Traits for Rootworm Control

### 3.1. RNAi as a New Mode of Action for Rootworm Control

Commercial traits used to control WCR have traditionally relied on insecticidal proteins identified from *B. thuringiensis* and other bacterial species [118,119,120]. When insecticidal proteins are expressed in transgenic maize, rootworm larvae exposed through root tissue feeding are significantly impacted in terms of mortality and stunting. Mortality is often the result of midgut epithelial cell disruption after receptor recognition results in surface membrane binding of the protein and a subsequent cascade of molecular events [121]. This effect in rootworm larvae translates directly into reduced root damage and yield protection.

As previously described, WCR is highly susceptible to environmental RNAi [122]. This has made it possible to target rootworm genes essential for survival through RNAi-mediated gene suppression as a new method to control rootworm damage complementary to insecticidal proteins. The next generation of rootworm-protected maize hybrids combine the MON87411 event, expressing Cry3Bb1 and *DvSnf7*, an RNAi trait, with the DAS-59122-7 event, expressing Cry34Ab1/Cry35Ab1, creating the first rootworm-active pyramid containing three modes of action [32]. A review of the efforts to identify efficient transgenic maize RNAi traits is provided below and a summary of the most successful genes evaluated in WCR is provided in Table 1.

### 3.2. Target Site Discovery

#### 3.2.1. Larvicidal RNAi

Given the successful deployment of transgenic rootworm control traits to date, a primary focus for developing RNAi as another mode of action (MOA) for WCR control has been directed at identification and validation of genes where suppression of gene expression leads to rapid cessation of larval feeding, stunting, and lethality. The aspiration is to develop an RNAi-based trait with performance comparable to existing protein-based traits in overall lethality and speed of kill. An RNAi-based MOA may result in overall mortality comparable to an insecticidal protein; however, generally, the kill speed is slower due to the intrinsic multistep cascade process that leads to gene suppression by RNAi [35,55,133,134]. Consequently, selecting target gene(s) is an essential factor determining whether an RNAi-based MOA can be successfully developed as a control trait. In WCR, the RNAi targets do not have to be limited to the insect midgut since WCR mounts a strong systemic response against dsRNA [56,113]. Ideal targets are those that are involved in critical physiological processes and, therefore, are essential to insect survival. Gene targets that have been characterized as effective candidates for larvicidal RNAi suppress essential biological functions including protein trafficking/sorting and transport (*snf7*, *sec23* [31,126]), organ integrity (*ssj1* [114]), energy metabolism (*v-ATPase* [55]), and muscle function (*troponin I* [127]) (Figure 1b).

Baum et al. [55] conducted the first large-scale screen of RNAi targets in WCR. A total of 290 dsRNA molecules were provided to WCR neonate larvae in diet feeding assays and numerous gene targets exhibited significant levels of mortality or growth inhibition. Among the most effective targets were putative orthologues for *vacuolar ATPase subunits A* (*v-ATPase A*) and *D* (*v-ATPase D)*, *COPI coatomer subunit β*, ribosomal proteins, and *ESCRT III snf7*. Ingestion of dsRNA targeting mRNA for these and other genes resulted in various levels of larval stunting and mortality. For *v-ATPase A*, a corresponding suppression of mRNA levels and generation of homologous siRNAs were demonstrated. v-ATPases are ATP-dependent proton pumps that function to transport protons across plasma membranes, acidify intracellular compartments, and play an essential role in membrane trafficking and protein degradation [55]. Significantly, the work by Baum et al. [55] provided the first demonstration of applying dsRNA technology for *in planta* insect resistance. Transgenic maize events expressing hairpin dsRNA targeting *v-ATPase A* conferred root protection against WCR feeding damage, demonstrating that artificial diet-based mortality could translate to transgenic plant-based reduction in root damage.

Further work on targets described by Baum et al. [55] led to the commercial development of dsRNA maize events targeting WCR *snf7* [31,124,125,135]. *DvSnf7* is a WCR ortholog of the *Drosophila vps32* or *shrub*, an essential component of the Endosomal Sorting Complex Required for Transport (ESCRT) pathway involved in intracellular protein trafficking [136]. Ultrastructural and histological studies showed that *snf7* suppression via exposure to *Dvsnf7* dsRNA in diet feeding assays led to progressive degeneration of midgut enterocytes, cell sloughing, cell lysis, and larval mortality [124,125]. RNAi-derived suppression of *snf7* represents the first RNAi-based MOA trait commercialized for insect pest control [32].

The successful engineering of *v-ATPase A* dsRNA for *in planta* control of WCR inspired RNAi target discovery research in WCR. Genome-wide screens utilizing high throughput target interrogation and knowledge-based approaches identified new RNAi-sensitive targets resulting in efficacious dsRNA maize events. Knorr et al. [123] tested 50 WCR genes selected based on genome-wide RNAi screens in *T. castaneum* [137]. From these, T0 maize plants expressing RNA hairpins for WCR orthologues *rop* (vesicular trafficking), *dre4* (transcription), or *rpII140* (transcription) showed protection from larval feeding damage. Knowledge-based approaches to dsRNA target selection have also been successful. Hu et al. [114] hypothesized, based on information from *Drosophila* genetic screens, that genes involved in insect midgut cell to cell septate junctions might be important to midgut integrity and function. Experimental evidence has shown that a class of WCR orthologous smooth septate junction (*ssj*) genes were sensitive to RNAi, resulting in insect mortality and plant protection. *DvSSJ1*, a homolog of the *snakeskin* (*ssk*) gene from *Drosophila*, prevents luminal content leakage across the midgut epithelial lining. Ultrastructural and histochemical studies performed by Hu et al. [114] showed that suppression of *DvSSJ1* destroys the integrity of SSJs and results in the collapse of midgut epithelial cells into the lumen, which disrupts midgut function, leading to feeding cessation and subsequent larval mortality [114,128]. Mortality directly correlated with a reduction in expression of *DvSSJ1* transcript and protein in WCR larvae.

Fishilevich et al. [127] examined a potentially haplolethal gene target, wings up A (*wup*A), which encodes Troponin I, an inhibitory protein of the Troponin-Tropomyosin complex involved in muscle contraction [138]. Due to the haplolethal phenotype, it was thought that RNAi of *wup*A might lead to greater dose sensitivity to dsRNA. RNAi of *wup*A resulted in a rapid onset of growth inhibition within two days. Knockdown of *wupA* resulted in significant food accumulation in the hindgut due to a loss of peristaltic motion of the alimentary canal. Finally, Vélez et al. [126] examined the lethal impact of dsRNA targeting *sec23*, a coatomer protein, and a component of the (COPII) complex that mediates ER-Golgi transport [139]. *sec23* was sensitive to RNAi in both larvae and adults. Interestingly, this study showed that 85% mRNA transcript knockdown resulted in 40% Sec23 protein knockdown, indicating that complete protein knockdown is not necessary to achieve insect mortality. Thus, it can be hypothesized that targeting important components of essential protein complexes could provide an effective route to insect control by disrupting protein complex stoichiometry. It should be noted that *sec23* dsRNA T0 maize events were protected against insect damage in greenhouse experiments and larval offspring of female adults that had been exposed to sublethal concentrations of *sec23* dsRNA in diet feeding assays displayed reduced weight and survival [126].

These examples illustrate that RNAi targets involved in essential processes can have larvicidal activity, translating to decreased root damage when hairpin RNAs are expressed. Although these genes are expressed throughout the larva, except for *DvSSJ1*, which is relegated to the midgut epithelia, it is unclear whether the larvicidal activity is primarily related to the loss of function of essential genes/gene products in the midgut/alimentary system cells compared to a more systemic effect on the larva.

#### 3.2.2. Parental and Reproductive RNAi

Naturally, most RNAi trait targets have focused on those that result in mortality due to the need to protect maize roots from damage by WCR larval feeding. However, given the susceptibility of WCR to environmental RNAi and the ability of RNAi-mediated gene suppression to spread systemically within WCR after ingestion, additional complementary approaches to WCR management can be explored. Parental RNAi is a new concept where adult oral exposure to dsRNA targeting embryonic development genes leads to effects on fecundity or effects within the offspring [115,130,131]. Two genes, *brahma* and *hunchback*, significantly impacted egg viability after female beetles were exposed to corresponding target gene dsRNA in diet feeding assays [115] (Figure 1b). *Brahma* (*DvBrm*) encodes an SWI/SNF ATP-dependent chromatin remodeler and *hunchback* (*DvHb*) encodes a zinc finger transcription factor [115]. Larvae from unhatched eggs laid by *DvHb* dsRNA-treated adult females primarily displayed deformation of abdominal and thoracic segmentation. This phenotype is consistent with the role of *DvHb* as an essential regulatory gene in the anterior–posterior patterning of insect embryos. No embryo development was observed from unhatched eggs laid by female beetles treated with *DvBrm* dsRNA. Additional chromatin remodeling ATPases including *chd-1*, *iswi-1*, *mi-2*, and *iswi2* were tested and generated a similar genotype to that observed with *DvBrm* [131]. These results demonstrated that adult exposure to dsRNA can manifest in their offspring [115,130,131]. However, the results obtained from diet-based assays did not translate effectively to *in planta* experiments with adults feeding on maize vegetative tissue expressing *DvBrm* or *DvHb* (data not published).

Niu et al. [132] targeted two reproductive genes, *DvVgr* and *DvBol*, to determine the impact of suppression on insect fecundity (Figure 1b). While oral feeding of *DvVgr* and *DvBol* dsRNA in diet-based assays using WCR larvae resulted in a reduction in fecundity, this trend did not translate to *in planta* results, a discrepancy possibly explained by the reduced expression of dsRNA found in transgenic plants compared to diet-based assays. A significant reduction in fecundity was only observed when WCR larvae fed on roots of transgenic maize plants expressing *DvBol* hairpin RNA. *bol* encodes an RNA-binding protein whose homolog in *Drosophila* impacts male spermatogenesis. Unexposed adult female WCR showed a significant reduction in egg-lay number and percentage of egg hatch after mating with adult male WCR that fed on transgenic maize roots expressing *DvBol* hairpin RNA. This resulted in an overall net reduction in fecundity between 84–95%. However, only a modest reduction in the number of eggs laid per female was observed from mated WCR that emerged after feeding on *DvVgr* hairpin RNA-expressing maize roots during the larval stage. These results reveal that RNAi targeting reproductive genes in WCR adults can persist throughout the development cycle after initial exposure as larvae and significantly impact adult fecundity. This phenomenon could be explained by maternally transmitted dsRNAs [140] or RNA-directed DNA or histone methylation [141]. However, these studies were performed in *T. castaneum* and *C. elegans* and *D. melanogaster*, respectively, and further exploration is necessary to unveil the specific mechanism allowing parental and reproductive RNAi in WCR.

The development of parental and reproductive RNAi as potential RNAi trait MOA represents a truly novel RNAi-based approach to mitigating WCR damage in the field by reducing or suppressing rootworm populations (Figure 1b). Combining different RNAi trait MOAs may be explored to manage rootworm populations further and slow the development of resistance to RNAi and the pyramided insecticidal proteins to protect yields from this devastating maize pest.

### 3.3. Transgenic dsRNA Events

Identification of gene targets that generated a robust RNAi response in WCR, and most importantly larval mortality, was the first step in developing insect-specific dsRNA as a management tactic. Incorporation of dsRNA molecules into transgenic maize events was the most viable option to ensure both stability of dsRNA and activity against WCR larvae in field environments. Deployment of transgenic maize hybrids with WCR-active dsRNA in field studies allowed researchers to understand the impact of the RNAi response on WCR survival and feeding injury and evaluate potential non-target effects in the field.

RNA molecules directed at target genes are carefully selected to avoid putative siRNA sequences with homology to transcripts from non-target organisms or the crop of interest, in this case, maize. Bioinformatic workflows are used to focus on regions of target transcripts predicted to generate a high abundance of siRNA. In WCR, dsRNA needs to be at least 60 bp in length to trigger gene knockdown effectively, and a length of 200–400 bases has shown to be an adequate size to generate a robust RNAi response in WCR. Midgut uptake of long hairpin dsRNA, but not siRNA 21-mers, is efficient in WCR [31]. A broad genome-wide uptake of endogenous plant dsRNA and transgenic dsRNA and subsequent processing of long dsRNA into 21-nucleotide (nt) siRNA by WCR has been demonstrated [142]. When developing RNAi traits, it is a routine practice to express inverted repeats homologous to the mRNA target from a strong constitutive promoter. Inverted repeats are separated by a non-homologous neutral stuffer sequence or intron. Such dsRNA expression constructs are exemplified by those described in Baum et al. [55], Armstrong et al. [143], and Hu et al. [129]. Transgenic maize events are selected when they contain only one or two expression construct copies to avoid silencing of the insert by the plant and simplify the event molecular characterization. Selected events for advancement are ultimately based on root protection against WCR feeding using a root damage rating system (0–3 node-injury scale) [144]. Further characterization of selected events examines the amount of dsRNA produced. This work is essential for determining non-target organism exposure levels in the field. dsRNA levels expressed in maize events are most efficiently quantified using QuantiGene RNA technology, a nucleic acid hybridization assay that can be run on crude maize tissue lysates without the need for extensive RNA purification as is required for reverse-transcription polymerase chain reaction quantitation [143]. Greenhouse and field-based testing of transgenic plants expressing *DvSnf7* indicated that by itself this RNAi-based trait MOA did not provide a level of protection comparable to a commercial trait, Cry3Bb1. However, combining both MOAs enhanced root protection and resulted in a significant reduction in WCR adult emergence, which may delay resistance development to both traits [32,145].

## 4. Field Efficacy of RNAi for Insect Control, Insect Resistance Management, and RNAi Resistance

### 4.1. Field Efficacy of RNAi Traits

The U.S. EPA approved the first RNAi product for insect control in 2017 [146], representing the first new MOA for WCR control since the release of the Cry34Ab1/Cry35Ab1 binary protein in 2005 [147]. As previously indicated, this product (SmartStax^®^ PRO) expresses three rootworm-active *Bt* proteins, Cry3Bb1 and Cry34Ab1/Cry35Ab1, as well as *DvSnf7* dsRNA [33]. SmartStax^®^ PRO also contains three Lepidopteran-active *Bt* proteins (Cry1A.105/Cry2Ab2 and Cry1F) and genes for glyphosate tolerance [33]. Given reports of field-evolved resistance to Cry3Bb1 [21,22,23,26,30], an efficacy evaluation of this product on Cry3Bb1-resistant insects and field populations was performed [148]. Additionally, *DvSnf7* dsRNA concentration-response larval bioassays were conducted on artificial diet using field-derived populations, a susceptible laboratory population, and a Cry3Bb1-resistant population. Lastly, greenhouse experiments evaluating beetle emergence from plants expressing each trait individually (i.e., Cry3Bb1, Cry34Ab1/Cry35Ab1, and *DvSnf7*) and in combination were conducted. The Cry3Bb1-resistant population exhibited a significant 2.7-fold decrease in susceptibility to *DvSnf7* dsRNA compared to the Cry3Bb1-susceptible population. However, the Cry3Bb1-resistant population lowered susceptibility was similar to other WCR field populations in diet bioassays. Cry3Bb1-resistant and susceptible colonies had similar WCR adult emergence from plants expressing *DvSnf7* and Cry34Ab1/Cry35Ab1, and WCR adult emergence from plants expressing *DvSnf7* and *DvSnf7* + Cry3Bb1 was not significantly different when the Cry3Bb1-resistant colony was evaluated [148]. Collectively, diet assay and *in planta* experiments have demonstrated a lack of cross-resistance between Cry3Bb1, Cry34Ab1/Cry35Ab1, and *DvSnf7* [148]. Previous research has also documented significant variation in susceptibility of WCR larvae from field populations to *DvSnf7,* with an LC_50_ ranging from 4.07 to 40.51 ng/cm^2^ [148]. This suggests that some populations might already exhibit higher tolerance to *DvSnf7* dsRNA and resistance monitoring will be essential to track susceptibility changes in field populations to promote trait durability.

Further field studies evaluated the efficacy of SmartStax^®^ (Cry3Bb1 + Cry34Ab1/35Ab1 pyramid) and SmartStax^®^ PRO maize (also expresses *DvSnf7* dsRNA) against western and northern corn rootworms [32]. Field trials conducted between 2013 and 2015 across the U.S. Corn Belt demonstrated that SmartStax^®^ PRO could significantly reduce root damage ratings under high WCR larval densities, in areas of Cry3Bb1 resistance, and in areas with greater than expected injury to Cry3Bb1 or SmartStax^®^. SmartStax^®^ PRO also significantly reduced root damage ratings relative to SmartStax^®^ in some field trials, indicating that *DvSnf7* dsRNA can provide additional root protection when coupled with *Bt* proteins [32]. In addition to enhanced root protection, SmartStax^®^ PRO also reduced adult emergence compared to SmartStax^®^, single-event Cry3Bb1 and Cry34Ab1/Cry35Ab1 hybrids, and the non-*Bt* control [32]. Collectively, the reduced root damage ratings and decreased adult emergence associated with SmartStax^®^ PRO suggest that the addition of *DvSnf7* and other RNAi traits could serve as a valuable tool for insect resistance management (IRM) strategies [32]. Various resistance modeling scenarios indicated that inclusion of *DvSnf7* dsRNA could promote durability of the expressed *Bt* proteins and decrease the rate of resistance evolution relative to SmartStax^®^, even in areas with suspected resistance to Cry3Bb1 [32]. However, no work has been performed to evaluate the role of RNAi traits under scenarios with resistance to both Cry3Bb1 and Cry34Ab1/Cry35Ab1. This product and other products with RNAi traits will likely play an important role in WCR population management, especially in locations with high annual WCR densities and confirmed resistance to Cry3Bb1 or Cry34Ab1/Cry35Ab1 [32].

### 4.2. Insect Resistance Management

The U.S. EPA requires that registrants of PIPs, such as *Bt* proteins and RNAi traits, complete and submit an IRM plan for the target pest before registration [149]. IRM is the scientific approach to delay the development of resistance in pest populations. SmartStax^®^ PRO follows the IRM pyramiding strategy and expresses three different MOAs targeting WCR (*DvSnf7* dsRNA, Cry3Bb1 and Cry34Ab1/Cry35Ab1), and stacks traits against Lepidopteran pests (Cry1F, Cry2Ab2, and Cry1A.105) and weeds (cp4/epsps (glyphosate resistance)). The IRM value of the pyramid is significantly reduced if individual components are deployed simultaneously, field-evolved resistance to one or more components is present, and/or cross-resistance between components is observed [150]. Confirmed WCR field-evolved resistance to Cry3Bb1 and/or Cry34Ab1/Cry35Ab1 has been reported in some populations [21,23,24,26,27,28,29,30], which could potentially impact the IRM value of this new pyramid. After this product is commercially available, it will be essential to monitor changes in susceptibility to the three traits, particularly Cry3Bb1 and Cry34Ab1/Cry35Ab1, to ensure durability of all traits in SmartStax^®^ PRO.

An important consideration for IRM with RNA-based traits is that, in contrast to *Bt* proteins, larvicidal dsRNAs can cause mortality in adult WCR [75,126,151,152]. The adult WCR RNAi response is rapid and can persist throughout most of the life stage. For example, knockdown of *Lac2* generated 76% knockdown 10 hours after ingestion and 86% knockdown 20 days after ingestion [153]. Expression of *DvSnf7* dsRNA occurs throughout the plant (event MON87411), including two tissues commonly consumed by adults in the field (e.g., pollen, leaves). However, the concentrations expressed *in planta* are not sufficient to generate mortality in adults and will provide sublethal exposure (mean of 0.103 ng/g and 33.8 ng/g in fresh weight pollen and leaf tissue, respectively [154]; the LC_50_ of *DvSnf7* previously observed in WCR adults was 60.2 ng/cm^2^ [151]). Therefore, adult sublethal exposure to *DvSnf7* may have implications for resistance management by adding selection pressure benefiting resistant individuals or individuals with resistance alleles [155,156]. Movement of adult WCR is common as maize phenology changes, with beetles searching for underdeveloped silks and pollinating maize as a primary food source [148]. Intrafield adult WCR movement could increase the likelihood of many individuals emerging from a single commercial maize field feeding on SmartStax^®^ PRO tissue at some point during the life cycle, while interfield movement of adult WCR and subsequent feeding on these plants would increase the risk of exposure to sublethal concentrations of dsRNA [157]. No fitness costs were observed in WCR adults exposed to *DvSec23* dsRNA LC_25_, suggesting that exposure to a sublethal concentration may not affect the fitness of exposed adults and their offspring [126]. However, due to the unique physiological effects of each dsRNA trait, future studies are important to determine the role sublethal exposure to *DvSnf7* field-relevant concentrations will have on adult WCR. 

Reports of *Bt* trait field-evolved resistance suggested that the high dose refuge (HDR) approach as a standalone IRM strategy was ineffective at delaying *Bt* resistance in WCR. The most recent draft of the U.S. EPA’s “Framework to Delay Corn Rootworm Resistance” requires incorporating integrated pest management (IPM) strategies into IRM plans to mitigate the spread of field-evolved resistance and minimize the risk of resistance evolution [158]. Suggested IPM strategies include crop rotation, rotation of PIP MOAs, adulticide application, and area-wide management [158]. Deploying this new dsRNA product within an IPM framework is necessary to increase trait durability and decrease the rate of resistance evolution [159].

### 4.3. Rootworm Resistance to RNAi

As in previous *Bt* PIPs, selection pressure from continuous exposure to dsRNA will eventually promote resistance evolution. Mechanisms of resistance to dsRNA were initially postulated to include degradation of dsRNA in the gut, reduced dsRNA uptake, alteration in proteins involved in dsRNA transport or formation of the RISC complex, loss of siRNA recognition by the RISC complex, mutation of the target gene, or systemic spread failure [76,134,160]. Identifying the resistance mechanism to *DvSnf7* dsRNA in WCR will inform effective IRM strategies to extend product durability if resistance is related to dsRNA processing, which may confer cross-resistance to other RNAi traits.

Efforts to evaluate the multigenerational effect of *DvSnf7* in field-collected insects resulted in the development of a resistant colony. Khajuria et al. [75] collected WCR adults emerging from areas planted with transgenic maize expressing *DvSnf7* dsRNA. Field-collected beetles were crossed with a non-diapausing WCR colony and exposed to *DvSnf7* dsRNA for eight generations, creating a population with ≥130-fold resistance to *DvSnf7* dsRNA [75]. Reciprocal crosses determined that *DvSnf7* dsRNA resistance was recessive, monogenic, and autosomal [75]. Resistance resulted from reduced uptake of dsRNA in gut cells and, interestingly, was found to be non-sequence specific. Cross-resistance to dsRNAs targeting *v-ATPase A*, *COPI B* (Coatomer Subunit beta), and *mov34* (26s proteasome) was identified, indicating adaptation occurred within the RNAi mechanism itself. Due to the development of cross-resistance to a variety of dsRNA targets in this study, it is possible dsRNA represents a single MOA in WCR [75]. However, it is yet unknown how insects will respond and adapt to dsRNA under field conditions. Further studies with other RNAi traits will provide a better understanding of potential resistance mechanisms that might exist in the field. 

Because *DvSnf7* will not be released as a single trait product, it was important to identify the chromosomal location of resistance alleles and compare with current knowledge on the location of resistance alleles to the *Bt* traits in SmartStax^®^ PRO. The Cry3Bb1 resistance gene(s) is located on linkage group 8 (LG8) [161] and the resistance locus for *DvSnf7* dsRNA is located on linkage group 4 (LG4) [75]. The fact that resistance causative elements are located on different chromosomes supports experimental evidence showing a lack of cross-resistance between *DvSnf7* dsRNA and Cry3Bb1 [75,148]. Therefore, resistance to both traits would have to develop independently and would have a lower probability of occurring than if a single trait or MOA were used.

Similar work in *L. decemlineata* furthers our understanding of how dsRNA resistance could develop in WCR. A laboratory-derived dsRNA-resistant *L. decemlineata* cell line and colony were recently established [105,162]. In both cases, cross-resistance to multiple dsRNAs was found, supporting results observed in WCR [75]. In dsRNA-resistant WCR, non-specific dsRNA uptake was disrupted, conferring resistance to all sequences tested. This fits with the transcriptional analysis of *L. decemlineata* cells, wherein genes related to uptake, *clathrin light chain*, *vha55*, *silA*, and the novel dsRNA binding protein *staufenC*, were downregulated in resistant cells [105]. However, while resistance in the WCR colony was narrowed to one locus in the genome, the *L. decemlineata* dsRNA-resistant colony displayed polygenic inheritance. This indicates that various mechanisms relating to dsRNA uptake could adapt to intense selection pressure. Given that resistance alleles have been detected in natural WCR field populations and those detected to date may confer cross-resistance to other dsRNA sequences, monitoring changes of susceptibility of field populations will be necessary to preserve RNAi technology. The deployment of *DvSnf7* with two additional functional MOAs will also assist in delaying RNAi trait resistance in populations that remain susceptible to Cry3Bb1 and Cry34Ab1/Cry35Ab1.

## 5. Environmental Risk Assessment

### 5.1. Environmental Risk Assessment Principles

Environmental Risk Assessment (ERA) is a three-step process that uses a science-based framework to evaluate whether a product of interest poses an unacceptable risk to the environment [163]. The framework is well established and broadly applicable to a wide array of substances [164], from traditional chemical molecules to traits used in the genetic modification of crops. The first step in the ERA process is problem formulation, which is a case-specific exercise that defines the scope and goals of the assessment, considers what is known about the substance, generates testable risk hypotheses in line with established protection goals, and formulates an approach for further data generation [165,166]. For example, the protection goals for non-target arthropods involve maintaining populations of beneficial arthropods that contribute to ecosystem services [167]. The most relevant populations evaluated include pollinators, parasitoids, and predators, as well as charismatic, protected, or endangered species. Once the species are selected, adequate surrogate species are selected to perform the testing in the next phase [168,169]. The flexibility and robustness of the ERA framework is driven by the case-specific nature of the problem formulation. Therefore, risk assessors must navigate the problem formulation phase before conducting thorough testing.

After problem formulation is complete, the analysis phase may begin. The analysis phase evaluates two key variables, exposure and hazard. The exposure assessment is conducted to identify routes of exposure between the substance of interest and valued non-target organisms and to characterize the magnitude and duration of such exposure [168]. The hazard characterization process evaluates the potency of the substance in a set of organisms from the problem formulation phase that serve as surrogate species for those that may be exposed in the environment [170,171]. Hazard or effects testing is often conducted in the context of a tiered framework, whereby early tier testing is conducted under highly conservative and controlled laboratory conditions [170]. For dsRNA, this can often be achieved by incorporating high concentrations of purified dsRNA, representing worse-case exposures, into an artificial diet to reach conservative test concentrations commonly used in Tier I testing. If no unacceptable effects are observed during this early tier testing, then the conclusion of minimal risk can be determined. If adverse effects are observed, higher tier tests may be conducted under more realistic conditions to determine whether adverse effects would likely occur under conditions more reflective of the environment [168]. The most common endpoint or effect evaluated is mortality, as it is unambiguous, easy to measure, and correlates with adverse effects to field populations. However, sublethal endpoints such as weight and developmental timing have also been used as additional endpoints in studies evaluating the risk of *Bt* crops (reviewed by Roberts et al. [172]). 

In the third phase of the ERA process, risk characterization, exposure, and hazard assessment results are combined and conclusions are drawn about the likelihood or magnitude of risk based on introducing the substance into the environment. Risk is ultimately a function of both exposure and hazard; thus, if either exposure or hazard is deemed minimal, then the risk associated with that substance is low [173].

### 5.2. Environmental Risk Assessment for RNAi Traits

The use of RNAi as a new MOA to confer a particular desired phenotype in a genetically modified crop has increased over the past decade [154,174]. Given the flexibility and robustness of the existing ERA framework described above, this structure is well suited to characterize the environmental safety of transgenic crops expressing dsRNA [172]. To briefly illustrate that process, we consider a dsRNA expressed in maize intended to control WCR.

An evaluation of possible exposure pathways is needed to focus the assessment on species that may be exposed to the introduced dsRNA sequence under conditions of use for the trait. This includes identifying which species may be present in maize fields, maize tissue types to which non-target organisms of interest may be exposed, and the concentrations of dsRNA in those tissues. Only those species that contribute to the identified protection goals, as explained in the previous section, need to be assessed. Conservative estimates of maize-expressed dsRNA exposure to non-target organisms are frequently based on high-end expression level determinations. These conservative estimates inform exposure levels of an organism, including detritivores and pollinators that feed directly on plant tissues. Plant dsRNA expression information can also estimate exposure to organisms in other environmental compartments, including soil and aquatic environments, using established models [163,175,176]. Soil and aquatic dissipation studies may be used if needed to refine exposure quantification and evaluate the potential for environmental persistence of dsRNA. Studies to date indicate that dsRNA dissipates rapidly and is unlikely to persist in soil and aquatic environments [177,178,179,180,181] (reviewed by Bachman et al. [182]). 

In addition to the exposure assessment, gathering baseline information about RNAi as a MOA to control WCR can help focus the risk assessment process. For instance, not all organisms appear to be sensitive to dsRNA, and those that are require sufficient sequence lengths and homology to the organismal mRNA for silencing to occur [31,129,183,184,185]. Once baseline information about general RNAi processes is gathered, understanding how conserved the gene mRNA sequence is among species related to WCR can provide testable hypotheses about the potential spectrum of activity for the introduced sequence [129]. By making use of ever-expanding bioinformatic databases and aligning known base pair sequence information, risk assessors can better identify which species may be sensitive to the introduced trait. An analysis plan can then be developed to incorporate these species, assuming they have relevant exposure pathways to the dsRNA, provide value to the risk assessment by contributing to the identified protection goals (i.e., pollination, predation, detritivory, etc.), and are available for testing in laboratory settings [171,186].

Developing an analysis plan for hazard testing involves an understanding of the ecosystem context for trait deployment (e.g., maize fields where WCR is present, hence non-target species in maize agroecosystems where WCR is present are considered). The analysis plan for hazard testing is also coupled with the exposure assessment (i.e., dsRNA concentration in relevant plant tissues and persistence in the environment) and the RNAi MOA and identified target gene, to generate hypotheses for which species may need to be tested in the laboratory. Increasing risk assessment certainty can be done by generating data to support or refute hypotheses on potentially sensitive species or characterizing the potential effects on organisms contributing to the identified protection goals. It is also important to consider endpoints or responses that are relevant to the risk assessment. Protection goals are typically framed around preserving ecosystem function based on the abundance of representative species. Therefore, endpoints such as mortality, weight, and time to development can be tied back to an effect on populations representing a specific ecosystem function and the overlying protection goals. A thorough understanding of the target gene is also helpful for designing studies and identifying relevant endpoints to observe in laboratory studies. The effects observed against the target pest, WCR, are informative for both the duration of laboratory studies and the endpoints that should be evaluated in those studies. Traditionally, mortality is the primary endpoint in risk assessment hazard studies with a threshold of at least 50% effect to trigger higher tier tests [168,172]. This is based on the notion that <50% effect at test concentrations in laboratory studies that exceed environmental concentrations are unlikely to result in adverse effects under field conditions. However, it should be noted that focusing on mortality for a non-essential gene may not provide relevant information and could lead to inaccurate conclusions on risk. In that way, understanding target gene function and the effects observed on the target pest from gene disruption can help risk assessors identify appropriate endpoints to observe in hazard studies. Similarly, careful observation of the timing of effects elicited on the target pest can help design laboratory hazard studies with beneficial insects important to protect. RNAi as a MOA relies upon protein suppression, a process that may take some time to lead to mortality. Therefore, it is important to ensure the duration of laboratory studies are long enough to measure the effect of the trait [154].

The case-specific and iterative nature of problem formation makes the ERA framework well suited to evaluate RNA-based traits. Collectively, the information gathered during problem formulation is used to generate specific and testable risk hypotheses relevant to a dsRNA-based trait that led to an analysis plan for exposure and hazard testing. Completing the analysis plan, detailing which data need to be generated, and producing those results mark the completion of problem formulation. Based on the analysis plan, data are generated during the analysis phase to characterize exposure and hazard posed by the trait of interest. After hazard and exposure data are collected, they are combined during risk characterization to ultimately make informed decisions about the likelihood the trait of interest poses an unacceptable risk to the environment or inform on problem formulation to guide additional testing.

### 5.3. Evaluation of Western Corn Rootworm RNAi Traits on Non-Target Organisms

Multiple studies have investigated the impact of WCR-active dsRNA constructs on non-target organisms. After the initial report that maize expressing *v-ATPase A* dsRNA could generate root protection from rootworm feeding damage, researchers started evaluating the potential effects of WCR *v-ATPase A* dsRNA on non-target arthropods, including a pollinator, *Apis mellifera* L. [187]; a herbivorous insect, *Danaus plexippus* (L.) larvae [188]; predators including the ladybeetle species, *Coccinella septempunctata* L., *Coleomegila maculata* (De Geer), *Hippodamia convergens* Guérin-Meneville, *Harmonia axyridis* (Pallas), and *Adalia bipunctata* (L.) [189,190]; and a decomposer, the collembolan *Sinella curviseta* Brook [191]. Results found that WCR *v-ATPase A* dsRNA has negligible effects on honey bee larvae and adults, monarch caterpillars, and *S. curviseta* [187,188,191]. Interestingly, species-specific *v-ATPase A* dsRNA experiments found that monarch larvae are not susceptible to dietary RNAi [188], and only a small downregulation from species-specific *v-ATPase A* dsRNA was observed in honey bees after 24 hours with gene expression recovering after 48 hours [187]. Furthermore, dsRNA was found bound to royal jelly, making it unavailable for honey bee larvae, suggesting environmental factors could render insecticidal dsRNAs unavailable for non-target organisms [187]. In contrast, two ladybeetles *H. axyridis* and *C. septempunctata,* experienced some gene knockdown and mortality when exposed to WCR *v-ATPase A* dsRNA [190]. In a different study, WCR *v-ATPase A* significantly reduced the development of *A. bipunctata* and *C. septempunctata* [189]. In these studies, ladybeetles with the strongest effects had more base pair matches between WCR *v-ATPase A* and their specific *v-ATPase A* sequence, highlighting the importance of bioinformatic analyses when designing insecticidal dsRNAs [189,190]. It is also important to consider that dsRNA concentrations tested in these studies were orders of magnitude higher than what is expected to occur in the field [189]; therefore, potential detrimental effects in a field scenario may be negligible.

Other studies have examined the impact of *DvSnf7* on non-target insects [154,192]. Bachman et al. [192] initially evaluated the lethal and sublethal effects of *DvSnf7* dsRNA on insects representing ten families and four orders. The *DvSnf7* dsRNA spectrum of activity was narrow, and activity was only observed in beetles within the Galerucinae subfamily of Chrysomelidae, whose homologous *snf7* sequence shared a 90% identity with WCR *Snf7* [192]. Tan et al. [193] evaluated *DvSnf7* dsRNA in honey bee larvae and adults and reported no adverse effects at high exposure levels. Bachman et al. [154] tested *DvSnf7* dsRNA activity at field concentrations in seven non-target insects, including honey bee larvae and adults; *C. maculata*; a rove beetle, *Aleochara bilineata* Gyllenhal; a ground beetle, *Poecilus chalcites* (Say); the green lacewing, *Chrysoperla carnea* (Stephens); the insidious flower bug, *Orius insidiosus* Say; and a parasitic wasp, *Pediobius foveolatus* (Crawford). Results from this study indicated that exposure to *DvSnf7* dsRNA at field-relevant exposure levels would not cause adverse effects on the non-target insect species described above [154]. These studies demonstrate the impact of *DvSnf7* dsRNA expressed in maize plants will be negligible on beneficial insects. Recent review papers provide more details regarding the problem formulation for off-target effects of dsRNA products for pest control [194], and sublethal endpoints for non-target organisms for insect-active transgenic crops, including RNAi [172].

## 6. RNAi Mammalian Safety

When evaluating human safety considerations for ingested RNA molecules associated with the use of RNA-mediated transgenic crops, one must consider the following: (1) the history of safe consumption of RNA; (2) biological barriers that limit internal exposure to exogenous RNAs; and (3) the weight of scientific evidence from published safety studies with RNA molecules. Collectively, these considerations enable proper hazard identification and, taken together with exposure assessment, can be leveraged to make an informed risk assessment decision on the use of transgenic maize expressing RNAi traits as a control tactic for corn rootworms.

### 6.1. RNA and Its History of Safe Consumption

RNAi is a ubiquitous process for gene expression modulation in all eukaryotic organisms. Therefore, longer dsRNA precursors and the small RNA molecules that this process leverages to regulate gene expression are also ubiquitous in widely consumed foods derived from plants and animals. Owing to the ubiquity of these RNA molecules, small RNAs with perfect sequence complementarity to transcripts of human and/or animal genes are evident in soybean, rice, maize, and fruits and vegetables [195,196,197]. These sequences include those important for key biological processes in mammals [196]. The presence of such sequences in staple food and feed crops demonstrates the safe consumption of small RNAs in the diet and supports the safety of RNAi for uses in transgenic crops, including those intended for control of agricultural pests such as corn rootworms.

Leveraging RNAi for insect control in a commercial transgenic crop represents a new application of this mechanism; however, RNAi has served as a natural process underlying plant phenotypes in domesticated crops and as a MOA in commercially approved transgenic traits (reviewed by Petrick et al. [198] and Sherman et al. [199]). One prominent example of RNA-mediated traits successfully leveraged in commercial transgenic products is the deployment of resistance to the papaya ringspot virus in papaya [200], a transgenic trait that played a vital role for Hawaiian growers amidst the devastation of the papaya crop. RNAi has also been leveraged commercially for the production of healthier oils in soybean [201,202], for reduction in browning in apples [203], and for reduction in acrylamide and blackspot bruising in potatoes [204,205]. Transgenic crops utilizing RNAi as a MOA have received regulatory approvals in several geographies. These include approvals for use in food; feed; and cultivation in crops such as maize, soybean, squash, potato, tomato, alfalfa, plum, apple, bean, and papaya [206]. The above information on RNA safe consumption in the diet and its safe use to date in commercialized transgenic crops with RNAi-based traits should also be applicable to those RNA-based traits intended for control of corn rootworms and other insects [55,129,145].

### 6.2. Biological Barriers to RNA Absorption

Owing to the ubiquitous presence of the RNAi mechanism in eukaryotic organisms, RNA molecules are ingested by vertebrates through the consumption of plants, animals, and fungi. Such RNA molecules include those with double-stranded regions that could initiate RNAi if they were to be absorbed from the diet and reach cells following consumption by the organism. As a further illustration of this point, exogenous dietary RNAs include those with sequence complementarity to vertebrate genes [195,196,197,207,208]. As vertebrates are constantly exposed to such RNA molecules, it is not surprising that there are a series of biological barriers that preclude these dietary components from serving in a regulatory capacity, and instead, they are harnessed for nutritional value. These biological barriers are reviewed in the scientific literature [198,199,209].

Ingested RNAs face an acidic digestive environment in the stomach following their initial exposure to nucleases in the saliva [210]. The efficacy of the low pH and hydrolysis in the stomach for nucleic acid degradation and the removal of bases from the nucleic acid backbone (e.g., depurination) has been well described and reviewed in the literature [198,209,211]. Further digestion of ingested nucleic acids occurs in the small intestine due to the presence of digestive enzymes and nucleases secreted by the pancreas [209]. Due to this extensive collection of barriers, nucleic acids from the diet are extensively degraded and do not undergo substantive absorption in an intact form, as demonstrated empirically with miRNAs in rodents [211,212], rhesus monkeys [213], and humans [214]. Thorough reviews of the safety considerations of plant expressed and externally applied RNA molecules for humans and other vertebrates have been published [198,215].

Biological barriers to the absorption and function of ingested nucleic acids expand beyond digestive barriers and include cellular membranes impermeable to RNAs that are both highly polar and large [198,216]. Each successive series of membrane barriers must be crossed for a dietary RNA to move from the intestinal tract lumen, across endothelial cells, and into the bloodstream. Once in the blood, RNAs are then subjected to nucleases [217,218,219] and rapid renal elimination [217,220]. To reach a putative target tissue, any RNA molecule escaping these nucleases would then have to cross through the endothelium and the epithelium within a target tissue to have the capacity to modulate tissue gene expression. Owing to the impermeability of these charged macromolecules across membrane barriers [198,216], any RNA surviving the intestinal tract would be unlikely to reach the bloodstream or a target tissue. Furthermore, RNAs reaching the cytoplasm of a cell in the consuming organism would subsequently be subjected to sequestration into endosomes that retain a vast majority of exogenous RNA molecules [221,222].

This array of physical, chemical, and biological barriers has made the development of pharmaceutical RNA drugs challenging, necessitating their chemical stabilization to limit degradation and specialized delivery formulations to elude barriers to exogenous RNA molecules [209,221,223,224]. Without such formulations, injection of these exogenous RNA molecules results in their rapid degradation and elimination [217,219,220,225]. Oral delivery of therapeutic macromolecules represents a desirable route of administration that is even more elusive to drug developers than systemically or locally administered therapeutics due to the aforementioned biological barriers [221,224]. These challenges are evident from literature reviews and studies demonstrating the need for specialized formulations and/or chemical modifications to facilitate limited delivery within proof-of-concept oral delivery evaluations of RNA therapeutics [209,223,226,227].

### 6.3. Evaluation of Potential Activity or Adverse Effects of Ingested RNA

The biological barriers described above collectively limit absorption of RNA molecules from the diet. Therefore, it is highly unlikely that ingested RNAs would have the capacity to regulate gene expression or induce adverse effects in a consuming mammalian organism. This concept stems primarily from a lack of significant RNA oral bioavailability [211,212], rendering the internal dose, e.g., the number of available copies of a given RNA molecule at the putative site of action, insufficient for regulatory function [214,219,228]. This is made especially challenging as uptake of RNA molecules in mammals is low (e.g., less than one copy per cell), and up to 1000 to 10,000 RNA copies per cell may be required for a functional RNAi response [229]. Further complicating the possibility of plant RNA activity in the diet is that small RNAs in plants are tightly bound to Argonaute proteins. Bound RNAs are not thought to dissociate from these complexes or exchange into Argonaute proteins in the consuming organism [228]. These principles collectively provide a solid biological basis for the history of safe dietary RNA consumption described above, even when ingested sequences possess sequence complementarity to transcripts in the consuming organisms.

The weight of evidence supporting the limited potential for functional activity ingested RNAs in mammals [230,231,232,233] has been called into question by several peer-reviewed publications alleging the opposite. However, a series of reviews and several laboratory studies on absorption and/or activity of ingested RNAs have been published since 2012, and collectively calls into question the potential for significant uptake and bioactivity of ingested RNAs [197,211,212,213,214,215,228,234,235,236,237]. For example, work challenging the concept of dietary RNA absorption/activity indicated that very low detection levels could have resulted from laboratory contamination or false-positive PCR results [213,238,239]. Furthermore, it is essential to ensure nutritionally balanced rodent diets when conducting rodent dietary studies. This is evidenced by one of the principal studies asserting uptake and activity of ingested RNAs in mammals [232]. Measured changes in blood cholesterol concomitant with dietary RNA detection in plasma were ultimately determined to result from dietary imbalances rather than ingested RNA activity [212]. The most comprehensive analysis of this phenomenon included an assessment of 800 human datasets, the results of which supported the conclusion that detection of small RNAs from exogenous sources in mammalian blood samples likely results from contamination as opposed to dietary uptake [235].

Toxicological evaluations of orally administered double-stranded RNAs have been conducted in mice to address the potential for oral activity and toxicity of these molecules. A 28-day repeated-dose oral toxicity study was conducted in mice with either a long dsRNA molecule or a pool of four 21 base pair siRNAs targeting the mouse *v-ATPase* gene. This proof of concept study for evaluating oral activity/toxicity of RNA was conducted utilizing a known gene target for corn rootworm control when expressed in plants [55] and to then construct a long dsRNA or a pool of predicted active siRNA sequences with 100% sequence complementarity to the mouse [237]. These mouse-targeting dsRNA sequences were then administered to mice by oral gavage for 28 consecutive days at doses of ≥48 mg/kg body weight, and traditional toxicology and target gene expression endpoints were evaluated. This study did not identify oral toxicity or suppression of *v-ATPase* gene expression in the gastrointestinal tract, kidney, liver, brain, or bone, demonstrating that biological barriers appear to preclude oral activity or toxicity of orally administered dsRNA molecules in mammals.

To further demonstrate RNA oral safety in a product-specific context, the *DvSnf7* dsRNA (240 base pair active dsRNA within a 968 nucleotide RNA sequence) was evaluated in a 28-day repeat-dose oral toxicity study [197]. *DvSnf7* dsRNA at oral doses of up to 100 mg/kg bodyweight for 28 consecutive days did not produce any treatment-related effects on weight, food consumption, clinical observations, clinical chemistry, hematology, gross pathology, or histopathology in mice [197]. Furthermore, the high dose utilized in this study (No-Observed Adverse-Effect-Level (NOAEL) of 100 mg/kg) was billions of times greater than highly conservative estimates of mean per capita human exposure to *DvSnf7* dsRNA in Europe, the U.S., Mexico, China, Japan, and Korea. This study demonstrates the safety of a dsRNA molecule used to control insects, specifically corn rootworms, illustrating further that transgenic crops using dsRNA for insect control do not pose unique risks to the health of consuming mammals, including humans.

## 7. Conclusions

The efficacy of dsRNA as a control tactic against WCR, along with commercial viability and regulatory approval, has spearheaded the use of RNAi technology for insect pest management and initiated a new phase of highly species-specific insecticides. The ERA and safety studies performed with dsRNA active against WCR have shown that the effects on non-target organisms, including humans and the environment, will be negligible.

The commercial release of dsRNA for corn rootworm control represents a milestone for WCR management, as it is the first new MOA to be deployed since 2005 [147]. However, the presence of dsRNA resistance alleles in natural populations exhibiting cross-resistance to other dsRNAs suggests RNAi may represent a single MOA in WCR. Further studies with other RNAi traits and WCR populations will determine how resistance may develop in the field and the uncertainty that presents for other RNAi traits. Furthermore, in contrast to historically commercialized *Bt* proteins, dsRNAs are more potent to adults at high concentrations, suggesting the likelihood of exposure to sublethal concentrations in the field upon feeding. Hence, future studies evaluating the effect of adult exposure to field-realistic dsRNA concentrations are important to characterize the potential risk of sublethal exposure on resistance evolution and the impact on WCR population dynamics. Resistance monitoring for the three MOAs expressed in SmartStax^®^ PRO and management of WCR within an IPM framework will be fundamental to guarantee the durability of these traits, ensuring available WCR control measures in upcoming years.

Finally, continued exploration into the WCR RNAi mechanisms (i.e., uptake gene silencing, and systemic spread) will benefit the search for putative routes of resistance, inform mitigation strategies, and perhaps lead to the development of efficient monitoring strategies utilizing bioinformatics and molecular approaches. In addition, an understanding of WCR RNAi mechanisms could provide insights into the efficiency (or lack thereof) in other insect pests, potentially providing paths for innovation. The development and validation of parental and reproductive RNAi have demonstrated the flexibility that RNAi could provide for insect pest management, representing a new approach to mitigating insect damage and a complementary approach to larvicidal RNAi.

The specificity of RNAi technologies continues to move agriculture towards more environmentally friendly insecticides. The lessons learned with WCR, the first insect to be targeted with an RNAi technology, can serve as an example for the development of novel RNAi insecticides in other insect pests, providing a safe and efficient MOA.

## Figures and Tables

**Figure 1 insects-13-00057-f001:**
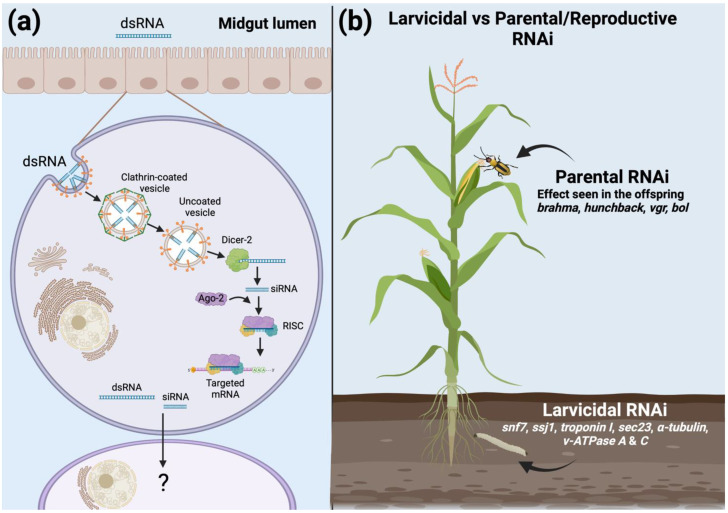
(**a**) Overview of the known RNAi uptake and processing mechanism in the western corn rootworm; (**b**) list of dsRNA gene targets active against the western corn rootworm subsequently transformed into maize.

**Table 1 insects-13-00057-t001:** RNAi genes orally evaluated in western corn rootworm. Larvicidal genes were tested in neonates and the evaluated phenotype was mortality. Larvicidal genes reported by Baum et al. [55] include those with an LC_50_ ≤ 5.2 ng/cm^2^, while genes from Knorr et al. [123] include those with ≤ 60% mortality in a nine-day bioassay. Parental and reproductive genes were tested in adults and the phenotype was evaluated in the offspring or adult fecundity, respectively.

Name	NCBI Accession No	Type of Exposure	Reference
**Larvicidal Genes**			
*ESCRT III snf7* ^1^	XM_028287710.1	Artificial diet, *in planta*	[31,55,124,125]
*v-ATPase-A* ^1^	XM_028281990	Artificial diet, *in planta*	[55]
*Actin* ^2^	XM_028292745.1	Artificial diet	[55]
*apple ATPase* ^1^	XM_028281191.1	Artificial diet	[55]
*alpha tubulin* ^1^	XM_028282553.1	Artificial diet	[55]
*beta tubulin* ^1^	XM_028282553.1	Artificial diet	[55]
*COPI coatomer subunit β* ^1^	XM_028291201.1	Artificial diet	[55]
*ESCRT III_vps2* ^1^	XM_028296669.1	Artificial diet	[55]
*ESCRT I-Vps28* ^1^	XM_028283797.1	Artificial diet	[55]
*mov34* ^1^	XM_028287237.1	Artificial diet	[55]
*ribosomal protein L9* ^1^	XM_028294395.1	Artificial diet	[55]
*ribosomal protein L19* ^1^	XM_028289442.1	Artificial diet	[55]
*ribosomal protein S4* ^1^	XM_028298505.1	Artificial diet	[55]
*ribosomal protein rps-14* ^2^	XM_028275245.1	Artificial diet	[55]
*RNA polymerase II* ^1^	XM_028297193.1	Artificial diet	[55]
*vATPase-D* ^1^	XM_028287428.1	Artificial diet	[55]
*sec23* ^1^	MK474471	Artificial diet, *in planta*	[126]
*wupA/troponin I* ^1^	MH001576.1	Artificial diet, *in planta*	[127]
*rop* ^3^	XM_028277045.1	Artificial diet, *in planta*	[123]
*dre4* ^3^	XM_028288745.1	Artificial diet, *in planta*	[123]
*rpIII 140* ^3^	XM_028297193.1	Artificial diet, *in planta*	[123]
*ncm-1*	XM_028276581.1	Artificial diet	[123]
*Rpb7-1*	XM_028299763.1	Artificial diet	[123]
*smooth septate junction protein 1 (ssj1)*	KU562965	Artificial diet, *in planta*	[114,128,129]
*smooth septate junction protein 1 (ssj2)*	KU562966	Artificial diet	[114]
*proteasome subunit beta type-1-like (protb)*	KU756279	Artificial diet	[114]
*proteasome subunit alpha type-3-like (pat3)*	KU756280	Artificial diet	[114]
*ribosomal protein S10 (rps10)*	KU756281	Artificial diet	[114]
**Parental and Reproductive Genes**			
*hunchback* ^1^	XM_028272853.1	Artificial diet, *in planta*	[115,130]
*brahma* ^1^	KR152260.1	Artificial diet, *in planta*	[115,130]
*chd-1* ^1^	KT364642	Artificial diet	[131]
*iswi-1* ^1^	KT364640	Artificial diet	[131]
*mi-2* ^1^	KT364639	Artificial diet	[131]
*iswi-2* ^1^	KT364641	Artificial diet	[131]
*Vgr* ^1^	KY373243	Artificial diet, *in planta*	[132]
*bol* ^1^	KY373244	Artificial diet, *in planta*	[132]

^1^ Orthologs identified from *D. melanogaster*; ^2^ Orthologs identified from *C. elegans*; ^3^ Orthologs identified from *T. castaneum.*

## Data Availability

No new data were created or analyzed in this study. Data sharing is not applicable to this article.

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
