# Peer review of "RNAi for Western Corn Rootworm Management: Lessons Learned, Challenges, and Future Directions"

_insects, 2022, doi:10.3390/insects13010057_

Round 1

Reviewer 1 Report

This article reviews the current state of knowledge about the use of RNAi approaches for controlling field populations of Diabrotica virgifera virgifera, the western corn rootworm (WCR). I think that the review is very exhaustive and transparent, bringing very significant insights on the subject, not only in WCR but in other pest insect/crop system. The paper is very well written for the most part, although somewhat heterogeneously, with some sections being more obscure and less refined (please see my comments below); I encourage the authors to polish these sections for increase readability. In any case, I am confident that this review will have a significant impact in the study of WCR and the field of agricultural entomology in general.

Author Response

We thank you and the Reviewers for a careful consideration of the manuscript

Minor/detailed comments:

L.72-92: These paragraphs may be better placed in introduction, especially as they are not specific 2 WCR and establish the scope of the review (ie. the focus on siRNAs only).

We understand your point of view, but believe the story makes sense as is. The intention of the introduction was to focus on WCR and bring in the new MOA (RNAi) at the end. Section 2 then focuses on RNAi and explaining what it is in the first sentence makes sense. Hopefully our response is sufficient. 

L.93-101: I think it would be nice to get more quantitative results from these studies, in order to be able to better frame the efficacy of the approach in WRC.

Statement was changed to reflect reviewer comments (additions italicized and underlined):

“In 2007, Baum et al. [55] demonstrated that oral exposure to DvV-ATPase A dsRNA elicited a silencing response in WCR larvae, leading to mortality (LC50 < 0.52 ng/cm2). In planta expression conferred maize root protection (mean node injury score of 0.25), exhibiting the possibility of utilizing the natural RNAi response for pest management. Another early study used RNAi in WCR to assess the function of WCR orthologues to laccase 2 (lac2) and chitin synthase 2 (chs2) by injecting dsRNA into second and third instars [56]. Injection of 200 ng/µl DvLac2-specific dsRNA resulted in a 95.8% reduction in lac2 expression and the prevention of post-molt cuticular tanning, while injection of 200 ng/µl DvCHS2-specific dsRNA reduced chitin levels in midgut tissues by 78%.”

L.114: Give some brief details about tag-130/chup-1 here (it is a bit obscure as is).

Added the statement “…tag-130/chup-1, a protein involved in cellular cholesterol import…” Tag-130/chup-1 is referenced in Line 128, providing more context.  

L.144: Please clarify what you mean by “due to speciation”.

“due to speciation” was changed to “due to physiological differences between species.” 

L.157-158: I think that this sentence is vague; please clarify.

Statement was altered to increase clarity based on reviewer comment:

“therefore, understanding how dsRNA enters and is released from the endosome in WCR may illustrate possible deficiencies in other insects.”

L.161: Please homogenize the format of “Dcr-2” throughout.

We used Dcr-2 when referring to the protein and its function and dcr-2 when referring to the gene, so we used the standard formatting used to mention proteins and genes.

L.169: Same for “R2D2”.

Similar to what we explained for Dcr-2, R2D2 refers to the protein and was used when referring to the function and r2d2 when referring to the gene. 

L.173-174: I would terminate the paragraph with this sentence.

Paragraph terminated after L 173-174 as suggested by reviewer.

L.205-206: Can you explain briefly why determining the role of crosstalk is important (in WRC and other models)?

The statement at Line 209 was amended to read:

“The extent to which crosstalk impacts the processing of exogenous dsRNA in WCR has not been determined. Further understanding of crosstalk between RNAi pathways may provide insight into why RNAi is efficient in Coleoptera and could be a cause of the differential response seen across Insecta.” 

Figure 1: 1b seems unnecessarily large. Given the information conveyed, I would rather see it as a sub-figure, about 1/3 the size of 1a.

While the comment is warranted, shrinking the image while retaining readability would be difficult. The editor agrees with this assessment to retain the image at its current size.

L.282: “control trait” instead of “trait”

“trait” was replaced with “control trait”

L.400: What do you mean by “like molecules”?

“like molecules” was replaced with “RNA molecules.”

L.405: What do you mean by “practical size range”?

Statement was changed to “ … a length of 200-400 bases has shown to be an adequate practical size range to generate a robust RNAi response in WCR for application.

L.400-427: This paragraph reads somewhat disjointly from the res of the text. The reasoning is not obvious and the point is not easy to understand.

An introduction was added to section 3.3. to improve flow, indicate the purpose of the section, and improve readability.

L.452-454: Is a differential “natural tolerance” among populations the only possible hypothesis? Wouldn’t the potential variance in treatment history or exposure to natural stressors play a role too?

Yes, similar hypotheses are being worked on now, however, nothing is published. Hopefully, to address your concerns, I removed the word “natural” from lines 462 and 464. The statement now reads:

“Previous research has also documented significant variation in susceptibility of WCR larvae from field populations to DvSnf7, with an LC50 ranging from 4.07 to 40.51 ng/cm2 [150]. This suggests that some populations might already exhibit higher tolerance to DvSnf7 dsRNA and resistance monitoring will be essential to track susceptibility changes in field populations to promote trait durability.”

By removing “natural” the exhibited variation could be due to a variety of things, but what they are is currently unknown.  

L.471: Do you mean “no work has been performed”?

“been” was added and the statement now reads “no work has been performed”

L.498-499: Change “LC50 of DvSnf7 in WCR adults = 60.2 498 ng/cm2 [150]), it constitutes” for “LC50 of DvSnf7 is 60.2 498 ng/cm2 in WCR adults [150]), which constitutes”

“=” was replaced with “is”; “it” was replaced with “which”.

L.500-502: Redundant reasoning (also, could be merged with previous sentence)

Merged with previous sentence. Statement now reads:

“However, the concentrations expressed in planta are not sufficient to generate mortality in adults and will provide sublethal exposure (mean of 0.103 ng/g and 33.8 ng/g in fresh weight pollen and leaf tissue, respectively [156]; the LC50 of DvSnf7 previously observed in WCR adults was 60.2 ng/cm2[153]). Therefore, adult sublethal exposure to DvSnf7 may have implications for resistance management by adding selection pressure benefiting resistant individuals or individuals with resistance alleles [157,158].”

L.508-512: Also redundant reasoning; could be simplified.

We believe the statement is important and decided to leave as is, but adjusted the rationale to indicate the importance.

“However, due to the unique physiological effects of each dsRNA trait, future studies are important to determine the role sublethal exposure to DvSnf7 field-relevant concentrations.”

L.524: “trigger” or “promote” instead of facilitate.

“facilitate” was changed to “promote”.

L.541-542: Ambiguous sentence

Sentence was altered for clarity:

“Due to the development of cross-resistance to a variety of dsRNA targets in this study, it is possible dsRNA represents a single MOA in WCR [75].”

L.605-606: Replace “Given the flexibility and robustness of the ERA framework described above, the existing framework” by “Given its flexibility and robustness, the existing ERA framework described above”

Statement was altered to reflect reviewer comment.

L.612: “to” instead of for”

“for” was changed to “to”. Further alteration of the sentence was carried out to improve clarity.

L.621: You mean “exposure quantification”?

            “quantification” was added after “exposure”.

L.620-623: There seem to be some contradiction between the 2 sentences. Please clarify.

The sentences do not seem contradictory to us. We first explained that exposure quantification studies MAY be used IF NEEDED to refine exposure. The next sentence shows that these studies have been done with dsRNA to determine potential persistence in the environment.

L.626: “appear to be sensitive”

Statement was changed to “appear to be sensitive.”

L.626-628: Grammar seems off.

Sentences were altered to improve clarity.

L.632-633: Sentence is vague. Please clarify/expand.

Agreed, sentence was deleted.

L.634-635: Or “risk assessors can better evaluate which species may be sensitive”

Statement was altered to reflect comment “risk assessors can better evaluate which species may be sensitive…”

L.363: You mean “exposure risk”?

*line 636 – added “pathways” after “exposure”.

L.641-642: I don’t understand the parenthesis

Statement in parenthesis altered for clarity based on reviewer comment:

“(e.g., in maize fields where WCR is present)”; deleted “are a driver for decision making”.

L.647: “relevant” and “meaningful” seem redundant here

“meaningful” was removed from statement.

L.648-649: The use of both function and services is ambiguous here.

“services” was eliminated from the sentence”.

L.649-651: very unclear/obscure

The sentence was rewritten for clarification. Evaluating mortality, weight, time of development or fecundity, to name a few allows to assess the potential effect on the populations representing a specific ecosystem function and the protection goals.

L.652: What do you mean by endpoint (in different places)?

We have provided a definition of endpoint, a common term used in ecological risk assessment, in section 5.1.

L.653: “nature of the effects”

“nature of the..” was eliminated.

L.640-668: I think this paragraph could be re-worked, as its reasoning is sometimes a bit obscure. 

The paragraph was adjusted for additional clarification.

L.679-679: This paragraph seems unnecessary because completely redundant with the previous one. Note: Overall, section 5.2 seems a bit disconnected from the rest and long for a part that does not relate to WCR directly.

We do not agree with removing this paragraph. Yes, it is a summary of the information above; however, it is a nice succinct synopsis of the information and it further reinforces one of the key questions we receive all the time in that is the current ERA process flexible and robust enough to adequately deal with non-Cry proteins. We obviously believe it is but that message, in my opinion, still needs to be reinforced.

L.760: The title of the section is not very informative

Title changed from “Biological Barriers” to “Biological Barriers to RNA Absorption”.

L.779: I think it is necessary to be a bit critical (briefly) of the fact that this demonstration has been made only in rodents.

We included an additional reference for a study performed in mice, one on rhesus monkeys, and one in humans. Furthermore, we guide the readers to two review papers with more thorough discussions of the topic.

L.807: Place “in a consuming organism” at the end of the sentence.

Sentence was altered to address comment. 

L.808-811: The link with previous sentences is not obvious.

The previous sentence (“Therefore, it is highly unlikely that ingested RNAs would have the capacity to regulate gene expression or induce adverse effects in a consuming mammalian organism.”).

L.824: Are you talking about the weak effects detected in those studies? Please clarify.

Entire paragraph was shuffled, and the comment addressed.  

L.825-837: This part could be streamlined a bit (esp. to remove loop-reasoning).

The entire paragraph was shuffled with the comment in mind. See lines 831 – 847.

L.873: “represents” instead of “is”

Statement was restructured to read: “However, the presence of dsRNA resistance alleles in natural populations exhibiting cross-resistance to other dsRNAs suggests that RNAi may represent a single MOA in WCR”.

L.877: The end of this sentence is odd.

Changed end of sentence to reflect route of adult exposure

L.879: You mean the risk of resistance evolution and the impact on the dynamics of WCR populations? Please clarify.

Statement was changed to: “Hence, future studies evaluating the effect of adult exposure to field-realistic dsRNA concentrations are important to characterize the potential risk of sublethal exposure on resistance evolution and the impact on WCR population dynamics.”

L.886: “approaches” instead of “techniques”

“techniques” was changed to “approaches”. 

L.887: “or lack thereof” in parenthesis

“or lack thereof” was changed to “(or lack thereof)”.

Reviewer 2 Report

The manuscript needs some revision, especially with respect to citations which are often not appropriate. Moreover it lacks consideration of possible future RNAi applications for corn rootworm control like exogenous dsRNA applications including stabilizing formulations.

Specific points of criticism:

  1. Introduction

L49-51: Crop rotation is mentioned as one of the control tactics for WCR. However, it seems unlikely that WCR has evolved resistance to all available management tactics including crop rotation. Therefore resistance should be limited to “all available chemical and biological plant protectants”.

L59: Ref. [26] needs to be deleted, because it is on resistance to other insecticides than Cry34Ab1/Cry35Ab1.

  1. RNAi in the Western Corn Rootworm

L118-121: This sentence is not very clear and needs to be specified, e.g. by adding that v-ATPase silencing was not affected at the transcript level and by specifying that silC knockdown had no effect on RNAi.

L137-139: Please check wording “knockdown of chc only slightly rescued silencing caused by a marker dsRNA”; presumably “rescued” is to be replaced by “reduced”.

L144-146: Specify what is meant by positive/negative results.

L187-188: The given reference is on human RISC. References on insect RISC should be included, also with regard to requirements for sequence matches (“almost perfect” is not precise).

L188-189: It is not conceivable why RNA hydrolysis and nuclease degradation are discriminated; ribonucleases in fact degrade RNA through hydrolysis.

L203-205: “Loquacious” should be specified as a dsRNA binding protein. However the term “miRNA” in brackets is misleading and should be deleted.

L216-218: The cited paper is on amplification of siRNA in a plant. This should be emphasized, or else a comparable report on viral immunity in insects needs to be referenced.

L234-235: It is stated here that siRNA or dsRNA travel cell to cell in coleopteran insects; this implies that not only dsRNA, but also siRNA is taken up by insect cells (compare to lines 102 and 109, which probably have to be amended).

  1. RNAi Traits for Rootworm Control

L245: Ref. no [124] cannot be correct here; please check

L308: The correct abbreviation is ESCRT

L379-381: Please consider possible explanations why oral feeding results did not translate to in planta results, e.g. because dsRNA is already processed in the plant, thus leading to reduced uptake.

L384-392: Please discuss possible explanations for parental RNAi, like transfer of siRNAs to the germline (although there is no indication for signal amplification in insects) or involvement of DNA methylation.

L408-410: Wording needs to be changed, i.e. “…. It is a routine practice to express inverted repeats homologous to the mRNA target from a strong constitutive promoter.

  1. Field Efficacy of RNAi for Insect Control, Insect Resistance Management and RNAi Resistance

L433-434: Please give more precise information on SmartStax PRO with regard to number of Bt toxins. There are obviously two Bt toxins specific for WCR, but there are also three additional Bt toxins with specificity to lepidopteran insects (see also lines 479-482).

L471: Word missing; it should be “has been performed”

L495-498: Specify the transgenic corn line for which expression data are given (SmartStax PRO?) Also, DvSnf7 expression in the initial event MON 87411 occurs throughout the plant (not just in two tissues) due to the constitutive e35S promoter. This information should be given here, because it is also relevant for environmental safety assessment. Therefore the wording needs to be adapted, e.g. Expression …. occurs in all plant parts, including tissues consumed by adults in the field (e.g. pollen, silks).

L518: Delete “resistance” after “minimize”

  1. Environmental Risk Assessment

L626-628: With respect to environmental risk assessment of RNAi for WCR control, not only data on length and homology requirements for WCR should be considered, but also for non-target organisms, which may be other insects, but also organisms from other groups; therefore more general references should be cited as well, e.g. the literature reviews by Paces et al. 2017 (EFSA Supporting publication 14(6), EN-1246) and by Christiaens et al. 2018 (EFSA Supporting Publication 15, EN-1424).

L711: Tan et al. [181] does not match ref. [181] in the reference list. Please check!

  1. RNAi Mammalian Safety

L731: Wrong wording “transgenic maize expression RNAi-traits as a control tactic”; may be changed into “transgenic maize expressing RNAi-traits as ….”

L745: Natural RNAi should not be considered as a technology; therefore better use the term “mechanism”.

L751-752: Inappropriate references; [196] is not on RNAi, [197) is not on RNAi in soybean. Alternative references need to be included, e.g. Wagner et al. (2011) Plant Biotechnol. J. 9, 723-728 and Yang et al. (2018) Transgenic Res. 27, 155-166.

L785: Ref. [211] is not appropriate regarding renal clearance. It may be substituted by Christensen et al. (2013) Drug Metabolism and Disposition 41, 1211-1219.

L813-815: While it is true that plant small RNAs are tightly bound to Argonaute proteins, this may not be true for small RNAs without homology to plant mRNAs, and it is not true for dsRNA which is the effective molecule taken up by the target insect. But there are other lines of evidence for the non-functionality of plant-derived dsRNA and siRNAs, e.g. the work of Chau and Lee (2007) Plant Methods 3, 13, who showed that siRNAs extracted from hairpin-expressing transgenic plants did not effectively silence mammalian genes.

References:

[150] Amendment needed: Journal of RNAi Gene Silencing 2016, 12, 528-535,

[166] and [169] Identical references! Reference source or link is missing.

[177] Correct to “… sediment-water system… ”

Author Response

We are grateful for the reviewer for pointing out reference mistakes and made the appropriate corrections. Regarding future RNAi applications, Dvssj1 maize is currently under evaluating for approval by the EPA, but that information cannot be included in the publication. In terms of exogenous dsRNA, this technology is not being considered and will not likely be considered for WCR management as using exogenous dsRNA applications to manage WCR adults would increase the likelihood of resistant evolution.

Line 49-51: Crop rotation is mentioned as one of the control tactics for WCR. However, it seems unlikely that WCR has evolved resistance to all available management tactics including crop rotation. Therefore resistance should be limited to “all available chemical and biological plant protectants

Rotation resistance has been identified in several areas of the Eastern U.S. Corn Belt (see [2,11]). These populations have circumvented crop rotation by laying eggs in non-host crops (e.g., soybean, oats, etc.), which then cause damage to first-year corn.

Line 59: Ref [26] needs to be deleted, because it is on resistance to other insecticides than Cy34Ab1Cry35Ab1

Suggested change made.

Line 118-121: this sentence is not very clear and needs to be specified, e.g. by adding that v-ATPase silencing was not affected at the transcript level and by specifying that silC knockdown had not effect on RNAi.

Statement was altered to reflect reviewer comment:

“However, in adult WCR, silA silencing did not affect V-ATPase knockdown after a secondary exposure to DvV-ATPase dsRNA. Despite this, mortality levels typically associated with a DvV-ATPase dsRNA exposure were reduced in silA knockdown treatments, while silencing of silC alone and in combination with silA had no effect on V-ATPase silencing or mortality.”

Line 137 – 139: please check wording “knockdown of chc only slightly rescued silencing caused by a market dsRNA”, presumably “rescued” is to be replaced by “reduced”

Removed “rescued” and replaced with “reduced”.

Line 144-146: Specify what is meant by positive/negative results

Statement was altered to reflect reviewer comment:

“Results suggesting involvement of silA and silC in the RNAi mechanism in T. castaneum, L. decemlineata, and WCR were found using the larval life stage. Similar experiments carried out with WCR adults provided less definitive conclusions.”

Line 187-189: The given reference is on human RISC. References on insect RISC should be included, also with regard to requirements for sequence matches (“almost perfect” is not precise) 

Reference on humans was removed. The included reference (formally [94], now [93]) covers RISC in Drosophila.

“almost perfect” was removed, and the following statement was added:

“Data from T. castaneum suggests a greater than 80% sequence identify with target mRNA is required for an efficient RNAi response and in D. melanogaster, 19 bp homology with mRNA led to effective target silencing”

With the following references:

  • Chen, J. et al. 2021. Off-target effects of RNAi correlate with the mismatch rate between dsRNA and non-target mRNA. RNA Biol. 18(11): 1747-1759
  • Kulkarni, M. M., et al. 2006. Evidence of off-target effects associated with long dsRNAs in Drosophila melanogaster cell-based assays. Nature Methods. 3: 833-838.

Line 188-189: it is not conceivable why RNA hydrolysis and nuclease degradation are discriminated; ribonucleases in fact degrade RNA through hydrolysis. 

According to the cited reference (Reference [100]), “The simplest and best understood consequence of target recognition is mRNA hydrolysis, or slicing, which can break the reading frame of the encoded protein and promote target degradation by cellular exonucleases.”

The statement at 188-189 was altered for clarity:

“After initial hydrolyzation of the mRNA target by Argonaute, the remaining nucleic acid is either degraded by endoribonucleases, exoribonucleases, or translated into incomplete proteins [96].”

Line 203-205: “Loquacious” should be specified as a dsRNA binding protein. However the term “miRNA” in brackets is misleading and should be deleted

Statement altered to address reviewer comment and clarify:

“the dsRNA binding protein Loquacious participates in both the siRNA and miRNA pathways in D. melanogaster, depending on the isoform expressed”.

Line 216-218: The cited paper is on amplification of siRNA in a plant. This should be emphasized or else a comparable report on viral immunity in insects needs to be referenced.

Reference [101] (plant) was replaced with

  • Karlikow, M., Goic, B., and M. C. Saleh. 2014. RNAi and antiviral defense in Drosophila: Setting up a systemic immune response. & Comparative Immunol. 42(1): 85-92
  • Tassetto, M., Kunitomi, M. and R. Andino. 2017. Circulating immune cells mediate a systemic RNAi-based adaptive antiviral response in Cell. 169(2): 314-325.e13

Line 234-235: It is stated here that siRNA or dsRNA travel cell to cell in coleopteran insects; this implies that not only dsRNA, but also siRNA is taken up by insect cells (compared to lines 102 and 109, which probably have to be amended).

Line 102 (“Silencing via the siRNA pathway includes three steps: dsRNA uptake, gene silencing, and systemic spread.”) refers to the environmental siRNA RNAi mechanism in a temporal sense, first long dsRNA must enter the cell, leading to silencing, followed by spread. Movement of siRNAs or dsRNAs referred to in lines 234-235, fall under “spread.” A clarification was added in the sentence to indicate that exosomes play a role in the secondary spread and not in the initial uptake in the gut. Publications have shown that in WCR especially, initial exposure to exogenous siRNA’s are not enough to elicit an RNAi response, therefore we begin with dsRNA uptake. Studies looking into  systemic spread in Drosophila and other coleopterans suggest that both dsRNA and siRNAs can be present in exosomes. However, this has not been evaluated in WCR.. It is our opinion that lines 102 and 109 clearly refer to the beginning of the process after exposure to exogenous dsRNA and lines 234-235 are clearly referring to the final step after the initial uptake and processing, the spread. Granted, parts of the mechanism seem to be quite circular after the initial exposure/cellular uptake of dsRNA. However, for clarity in explaining the mode of action in the introductory sentence (line 102), we chose to define an ending.

Line 245: Ref. no [124] cannot be correct here; please check

Reference was changed to

Dominguez-Arrizabalga et al. (2020). Insecticidal activity of Bacillus thurningiensis proteins against coleopteran pests. Toxins (Basel) 12.

Line 308: the correct abbreviation is ESCRT

“ESCORT” was changed to “ESCRT”. 

Line 379-381: Please consider possible explanations why oral feeding results did not translate to in planta results. e.g. because dsRNA is already processed in the plant, thus leading to reduced uptake

Sentence was augmented to reflect reviewer comment:

“While oral feeding of DvVgr and DvBol dsRNA in diet-based assays using WCR larvae resulted in a reduction in fecundity, this trend did not translate to in planta results, a discrepancy possibly explained by the reduced expression of dsRNA found in transgenic plants compared to diet-based assays”.

Line 384-392: Please discuss the possible explanations for parental RNAi, like transfer of siRNAs to the germline (although there is no indication for signal amplification in insects) or involvement of DNA methylation

We provided two possible explanations for the parental RNAi response and provided references for the potential mechanism.

Line 408-410: wording needs to be changed, i.e. “…it is a routine practice to express inverted repeats homologous to the mRNA target from a strong constitutive promoter”

Suggested change made.

Line 433-434 Please give more precise information on SmartStax PRO with regard to number of Bt Insecticidal Activity of Bacillus thuringiensis Proteins against Coleopteran Pests toxins. There are obviously two Bt toxins specific for WCR, but there are also three additional Bt toxins with specificity to lepidoptera insects (see also lines 479-482)

As previously indicated, this product (SmartStax® PRO) expresses three rootworm-active Bt toxins, Cry3Bb1 and Cry34Ab1/Cry35Ab1, as well as DvSnf7 dsRNA [32]. SmartStax® PRO also contains three Lepidopteran-active Bt toxins (Cry1A.105/Cry2Ab2 and Cry1F) and genes for glyphosate tolerance.

Line 471: Word missing; it should be “has been performed”

Suggested change made.

Line 495-498: Specify the transgenic corn line for which expression data are given (SmartStax PRO?) Also, DvSnf7 expression in the initial event MON 87411 occurs throughout the plant (not just in two tissues) due to the constitutive e35S promoter. This information should be given here, because it is also relevant for environmental safety assessment. Therefore the wording needs to be adapted, e.g. Expression … occurs in all plant parts, including tissues consumed by adults in the field (e.g. pollen, silks).

"MON 87411” was added to:

“Expression of DvSnf7 dsRNA occurs throughout the plant (event MON87411), including two tissues commonly consumed by adults in the field (e.g., pollen, silks).”

“Throughout the plant” was added to:

“Expression of DvSnf7 dsRNA occurs throughout the plant, including two tissues commonly consumed by adults in the field (e.g., pollen, silks).”

Line 518: delete “resistance” after “minimize”

Suggested change made.

Line 626-628: With respect to environmental risk assessment of RNAi for WCR control, not only data on length and homology requirements should be considered, but also for non-target organisms which may be other insects, but also organisms from other groups; therefore more general references should be cited as well, e.g. the literature reviews by Paces et al. 2017 (EFSA Supporting publication 14(6) and by Christiaens et al. 2018 (EFSA Supporting Publication 15, EN-1424).

Paces et al. and Christiaens et al. were added as citations. We appreciate the reviewer bringing these references to our attention.

Line 711 Tan et al [181] does not match ref. [181] in the reference list. Please check!

Reference was fixed.

Line731: wrong wording “transgenic maize expression RNAI-traits as a control tactic”, may be changed into “transgenic maize expression RNAi traits as…”

Changed “expression” to “expressing”.

Line 745: Natural RNAi should not be considered as a technology; therefore better use the term mechanism

Technology” was changed to “mechanism”.

Line 751-752: Inappropriate references [196] is not RNAi, [197] is not on RNAi in soybean, Alternative references needed to be included, e.g. Wagner et al. (2011) Plant Biotechnol. J. 9, 723-728 and Yang et al. (2018) Transgenic Res. 27 155-166.

References were changed to Wagner et al. and Yang et al. We appreciate the reviewer bringing these to our attention. 

Line 785: Ref [211] is not appropriate regarding renal clearance. It may be substituted by Christensen et al. (2013) Drug Metabolism and Disposition 41, 1211-1219

Reference [211] was changed to Christensen et al. (now Ref. [222])

Line 813-815: While it is true that plant small RNAs are tightly bound to Argonaute proteins, this may not be true for small RNAs without homology to plant mRNAs, and it is not true for dsRNA which is the effective molecule taken up by the target insect. But there are other lines of evidence for the non-functionality of plant-derived dsRNA and siRNAs, e.g. the work of Chau and Lee (2007) Plant Methods 3, 13, who showed that siRNAs extracted from hairpin-expressing transgenic plants did not effectively silence mammalian genes.  

The point of our argument around Argonaute proteins is that it further explains the safety of plant expressed RNAs in nature despite mammalian homology, but not for transgenes per se. This is clear in the last statement of the paragraph in question in that it indicates the biological basis in part for safety of dietary RNA consumption from plants if one were to raise concerns about their ability to regulate gene expression after consumption. Agreed, there is not certainly of this mechanism with respect to those from transgenes that don’t have homology to the plant. In insects, the work from Bachman et al and Baum et al and the broader body of work published by Monsanto and others demonstrate that it is the longer dsRNAs and not siRNAs from the transgene that get taken up into the insect and then mediate silencing through the insect DICER/RNAi pathway.  I believe that they must be >60bp for uptake and activity. While the work of Chau and Lee is interesting, there is no logical reason to believe this would hold across all siRNAs expressed transgenically in a plant. If the siRNA were to match a mammalian gene and were dosed in sufficient quantity to mammalian cells with a transfection reagent, such an artificial in vitro experiment could produce gene suppression.  The biological barriers to such an RNA actually reaching a mammalian cell in meaningful quantities to produce even a change in gene expression after oral ingestion of plant material is really the key issue here and that is the crux of the provided argumentation. While interesting, Chau and Lee may be an outlier and we prefer not to cite this as evidence that plant expressed siRNAs don’t work in an in vitro system.

References:

[150] Amendment needed: Journal of RNAi gene silencing 2016, 12, 528-535

Correction was made to reference.

[166] and [169] identical references! Reference source or link is missing

Ref [169] was removed and re-cited as [163].

[177] correct to “…sediment-water system…”

Correction was made to reference.

Reviewer 3 Report

Darlington et al. have compiled an encyclopedic review of RNAi research pertaining to the western corn rootworm. Very little was excluded from this comprehensive review. The authors synthesize the entirety of the rootworm literature on this topic into sections containing highly detailed explanations of published research bringing the field fully up to date. The subject was covered from RNAi delivery, mechanisms of action, evolution of resistance and resistance managemtn, to digestion and environmental effects on nontargets, and regulatory concerns. Even a section on the controversial claims of eRNA consumption was included. Perhaps the only subjects left wanting were the engineering of proper expression in plants, which would likely warrant an even longer review, and comparisons with the various more stable structures and forms of RNA, the targeting of splice sites, and encapsulation technologies which have been tested on other insect species. But the authors endeavored to limit their ambit to the rootworm literature. This review once published will certainly be cited by many in the field.

Author Response

The authors thank this reviewer for their comments. We agree that this review will be a widely-read and published article with important knowledge of WCR RNAi. We agree with the reviewer that including a section on engineering of proper expression in plants, different forms of dsRNA, the different splice sites and encapsulation technologies are of interest. However, they are out of the scope of the review as we wanted to focus on the current  knowledge for WCR.

Round 2

Reviewer 2 Report

The manuscript has been improved considerably. There are only some small corrections that need to be made:

L 407: Histone methylation and DNA methylation are two distinct epigenetic modifications involved in regulation of gene expression. Therefore the sentence has to be changed to ... histone and DNA methylation ....

Reference 153 (Pereira et al.): Reference is still incomplete. Please check the correct name of the journal.

Author Response

Dear reviewer,

We appreciate your time to review the manuscript for a second time. Below are the corrections to the suggestions provided.

L 407: Histone methylation and DNA methylation are two distinct epigenetic modifications involved in regulation of gene expression. Therefore the sentence has to be changed to ... histone and DNA methylation ....

A couple of portions of text from these paragraphs, including the text mentioned by the reviewer above, were revised to increase clarity. 

Reference 153 (Pereira et al.): Reference is still incomplete. Please check the correct name of the journal.

J RNAi Gene Silencing and page numbers added. Thank you for catching this error.

Sincerely,

Ana M. Vélez

Assistant Professor

University of Nebrask-Lincoln